# The tectonigral pathway regulates appetitive locomotion in predatory hunting in mice

Meizhu Huang[1,10], Dapeng Li [2,10,11✉], Xinyu Cheng[3,4,10], Qing Pei[4,10], Zhiyong Xie[4,10], Huating Gu[4], Xuerong Zhang[4], Zijun Chen[5], Aixue Liu[3,4], Yi Wang [5], Fangmiao Sun[6], Yulong Li [6], Jiayi Zhang [7], Miao He [7], Yuan Xie[8], Fan Zhang[8], Xiangbing Qi [4,9], Congping Shang[1,11✉] & Peng Cao [4,9,11✉]

Appetitive locomotion is essential for animals to approach rewards, such as food and prey. The neuronal circuitry controlling appetitive locomotion is unclear. In a goal-directed behavior—predatory hunting, we show an excitatory brain circuit from the superior colliculus (SC) to the substantia nigra pars compacta (SNc) to enhance appetitive locomotion in mice. This tectonigral pathway transmits locomotion-speed signals to dopamine neurons and triggers dopamine release in the dorsal striatum. Synaptic inactivation of this pathway impairs appetitive locomotion but not defensive locomotion. Conversely, activation of this pathway increases the speed and frequency of approach during predatory hunting, an effect that depends on the activities of SNc dopamine neurons. Together, these data reveal that the SC regulates locomotion-speed signals to SNc dopamine neurons to enhance appetitive locomotion in mice.

[1] Bioland Laboratory (Guangzhou Regenerative Medicine and Health Guangdong Laboratory), Guangzhou, China. [2] Department of Neurobiology, School of Basic Medical Sciences, Beijing Key Laboratory of Neural Regeneration and Repair, Advanced Innovation Center for Human Brain Protection, Capital Medical University, Beijing, China. [3] Graduate School of Peking Union Medical College, Chinese Academy of Medical Sciences, Beijing, China. [4] National Institute of Biological Sciences, Beijing, China. [5] State Key Laboratory of Brain and Cognitive Sciences, Institute of Biophysics, Chinese Academy of Sciences, Beijing, China. [6] College of Life Sciences, Peking University, Beijing, China. [7] State Key Laboratory of Medical Neurobiology, Fudan University, Shanghai, China. [8] Key Laboratory of Neural and Vascular Biology in Ministry of Education, Department of Pharmacology, Hebei Medical University, Shijiazhuang, Hebei, China. [9] Tsinghua Institute of Multidisciplinary Biomedical Research, Tsinghua University, Beijing, China. [10] These authors contributed equally: Meizhu Huang, Dapeng Li, Xinyu Cheng, Qing Pei, Zhiyong Xie. [11] These authors jointly supervised this work. Dapeng Li, Congping Shang, Peng Cao. ✉email: lidapeng2021@ccmu.edu.cn; shang_congping@grmh-gdl.cn; caopeng@nibs.ac.cn

L ocomotion plays a fundamental role in the survival of organisms. It can be conceptually divided into three categories (i.e., appetitive locomotion, defensive locomotion, and exploratory locomotion)[1]. These three types of locomotion are selectively used by organisms for specific behavioral needs[2]. Appetitive locomotion is indispensable for organisms to approach rewarding targets. For example, in a naturalistic goal-directed behavior—predatory hunting, predators employ appetitive locomotion to chase and catch up with prey[3,4]. How the brain controls appetitive locomotion during predatory hunting is an unresolved question in the field of neuroethology[5].

The superior colliculus (SC) is a multi-layered midbrain structure for sensory information processing and motor functions[6–8]. The superficial layers of the SC primarily receive visual inputs[9] and perform visual information processing[10,11]. The intermediate and deep layers of the SC are involved in sensorimotor transformation and motor functions[7]. The motor functions of the SC include saccadic eye movement[12–14], head movement[15,16], and locomotion[17,18]. From a neuroethological perspective, the SC may use these motor functions to orchestrate distinct behavioral actions in predatory hunting in rodents[19–22].

How the SC orchestrates distinct behavioral actions during predatory hunting (e.g., approaching and attacking prey) is beginning to be elucidated. With an unbiased activity-dependent genetic labeling approach (FosTRAP2), several hunting-associated tectofugal pathways were identified, such as those projecting to the zona incerta (ZI) and the substantia nigra pars compacta (SNc)[22]. While the SC–ZI pathway is primarily involved in sensory-triggered predatory attack during hunting, the functional role of the SC–SNc pathway in predatory hunting has not been revealed yet.

The SC–SNc pathway, also known as tectonigral pathway, was first described by Comoli et al. (2003). It was shown that neurons in the intermediate and deep layers of the SC form synaptic contacts with dopamine and non-dopamine SNc neurons[23]. Considering the recent studies showing the involvement of SNc dopamine neurons in the vigor of body movements, including locomotion[24–28], we hypothesized that the SC–SNc pathway may participate in appetitive locomotion during predatory hunting.

In the present study, we explored the role of SC–SNc pathway in appetitive locomotion during predatory hunting. We found that the SC–SNc pathway transmitted locomotion speed signals to SNc dopamine neurons and triggered dopamine release in the dorsal striatum. Activation of this pathway during predatory hunting increased the speed of appetitive locomotion, an effect that depended on the activities of SNc dopamine neurons. Conversely, synaptic inactivation of this pathway impaired appetitive locomotion without changing defensive locomotion. Together, these data revealed the SC as an important source to provide locomotion-related signals to SNc dopamine neurons to boost appetitive locomotion.

## Results

**The SC–SNc pathway is primarily glutamatergic.** We began this study by performing morphological analyses of the SC–SNc pathway. First, we mapped the SC–SNc pathway with cell-type-specific expression of "SynaptoTag"[29], which is the enhanced green fluorescent protein fused to synaptic vesicle protein synaptobrevin-2 (EGFP-Syb2). AAV-DIO-EGFP-Syb2 was unilaterally injected into the SC of vGlut2-IRES-Cre or GAD2-IRES-Cre mice (Fig. 1a, c). The specificities of vGlut2-IRES-Cre and GAD2-IRES-Cre mice to label glutamate+ and GABA+ SC neurons have been validated in an earlier study[22]. EGFP-Syb2 expression in SC neurons of vGlut2-IRES-Cre mice resulted in considerable EGFP-Syb2+ puncta in the SNc (Fig. 1b and Supplementary Fig. 1a). In contrast,

sparse EGFP-Syb2+ puncta were observed in the SNc of GAD2-IRES-Cre mice (Fig. 1d and Supplementary Fig. 1b). We normalized the density of EGFP-Syb2 puncta by dividing the puncta density in the SNc with that in the SC of each mouse. Strikingly, the normalized density of EGFP-Syb2 puncta in the SNc of vGlut2-IRES-Cre mice was significantly higher than that of GAD2-IRES-Cre mice, suggesting that the SC–SNc pathway is primarily glutamatergic (Fig. 1e).

Second, we retrogradely labeled SNc-projecting SC neurons by injecting CTB-555 into the SNc of WT mice (Fig. 1f). The retrogradely labeled cells (CTB-555+) in the SC were distributed predominantly in the intermediate and deep layers (Fig. 1g and Supplementary Fig. 1c). By using primary antibodies that specifically recognize GABA and glutamate, we found that most of the CTB-555+ cells were immunohistochemically glutamate-positive (92 ± 3%, $n = 5$ mice; Fig. 1h) and GABA-negative (91 ± 3%, $n = 5$ mice; Fig. 1i). These data, again, suggested that the SC–SNc pathway is primarily glutamatergic.

**The SC–SNc pathway is anatomically segregated from other tectofugal pathways.** We wondered how the SC–SNc pathway is anatomically related to other tectofugal pathways. Some SC neurons monosynaptically innervate the ventral tegmental area (VTA) and were implicated in the regulation of wakefulness and innate defensive responses[30,31]. To test the anatomical relationship between the SC–SNc and SC–VTA pathways, we injected CTB-488 and CTB-555 into the SNc and VTA of the same WT mice (Supplementary Fig. 2a). In the SC sections, only a few cells were dually labeled by CTB-488 and CTB-555 (Supplementary Fig. 2b). Quantitative analyses of CTB-labeled cells indicated that 11.3% SNc-projecting SC neurons innervate the VTA, whereas 9.7% VTA-projecting SC neurons innervate the SNc (Supplementary Fig. 2i). Then we tested whether the SNc-projecting SC neurons send collaterals to the ZI, an important center for feeding-related predation[32,33]. Injection of CTB-488 and CTB-555 into the SNc and ZI of the same mice (Supplementary Fig. 2c) resulted in only a few dually labeled cells in the SC sections (Supplementary Fig. 2d). Quantitative analyses indicated that 9.8% SNc-projecting SC neurons innervate the ZI, whereas 11.1% ZI-projecting SC neurons innervate the SNc (Supplementary Fig. 2j). We also examined the anatomical relationship between the SC–SNc and SC–LPTN pathways, the latter of which is involved in innate fear response to overhead looming visual stimuli[34,35]. We found that injection of CTB-488 and CTB-555 into the SNc and LPTN resulted in very few dually labeled SC neurons (Supplementary Fig. 2e, f). Quantitative analyses indicated that 6.2% SNc-projecting SC neurons innervate the LPTN, whereas 5.2% LPTN-projecting SC neurons innervate the SNc (Supplementary Fig. 2k). As a control experiment, injection of mixed CTB-488 and CTB-555 in the SNc (Supplementary Fig. 2g) resulted in SC neurons co-labeled by both CTB-488 and CTB-555 (Supplementary Fig. 2h, l). Together, these data suggest that the SC–SNc pathway is largely segregated from the SC–VTA, SC–ZI, and SC–LPTN pathways in mice.

**The SC–SNc pathway preferentially innervates SNc dopamine neurons.** To explore how the SC–SNc pathway is synaptically connected to the SNc neurons, we employed mouse lines to genetically label different neuronal subtypes in the SNc. The most prominent neuronal subtype in the SNc is the dopamine neurons positive for tyrosine hydroxylase (TH+). These SNc dopamine neurons are largely segregated from those expressing glutamate decarboxylase-2 (GAD2+)[36,37] or vesicular glutamate transporter-2 (vGlut2+)[38,39]. Although TH-GFP mice[40] did not reliably label dopamine neurons in the VTA, this line marked

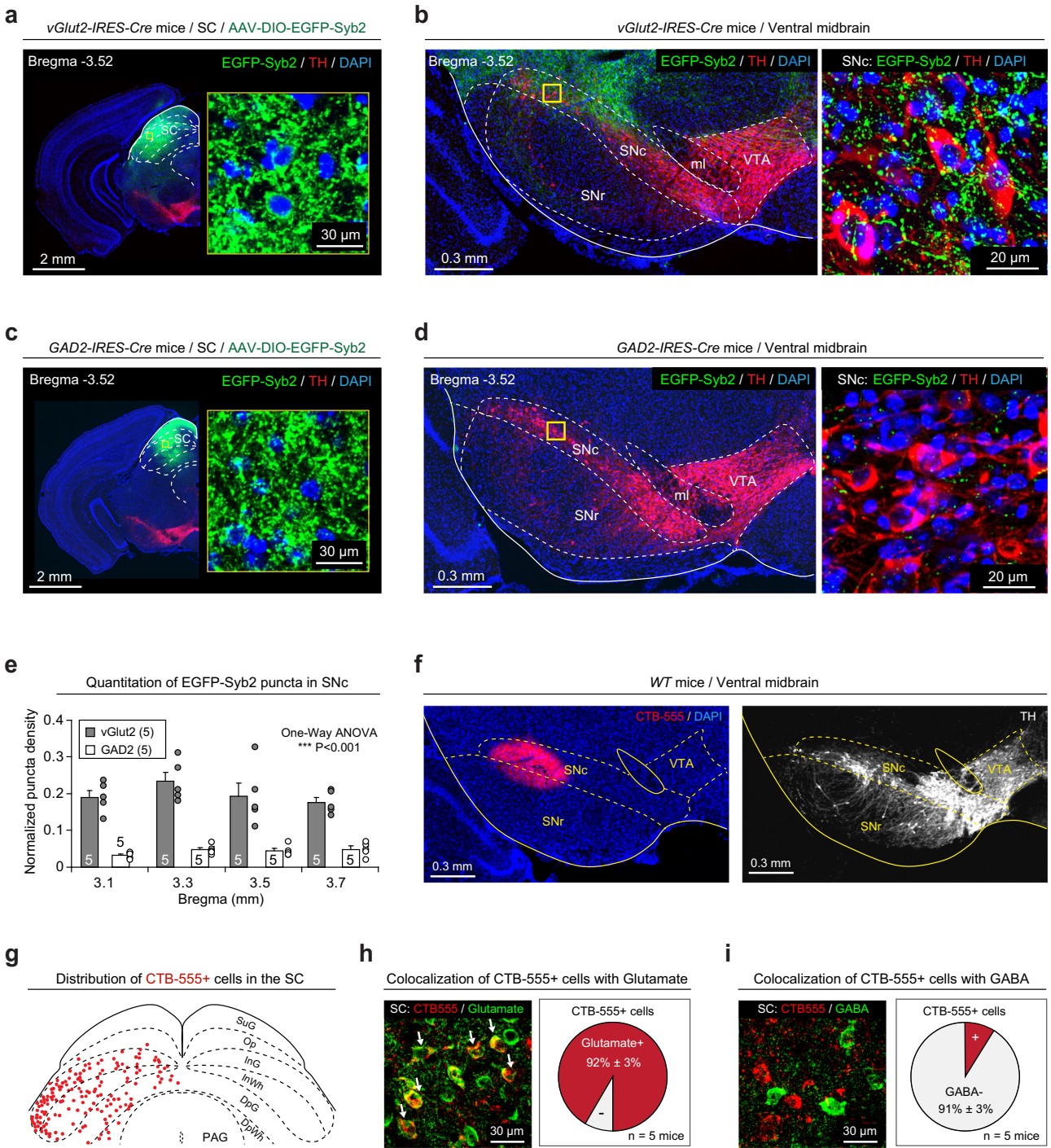

**Fig. 1 Cell-type-specific mapping of the SC–SNc pathway. a, c** Example coronal brain sections of vGlut2-IRES-Cre (**a**) and GAD2-IRES-Cre mice (**c**) with EGFP-Syb2 expression in the SC. Insets, high-magnification micrographs showing EGFP-Syb2+ puncta in the SC. **b, d** Left, EGFP-Syb2+ axon terminals in the ventral midbrain of vGlut2-IRES-Cre (**b**) or GAD2-IRES-Cre mice (**d**). Right, high-magnification micrographs showing EGFP-Syb2+ puncta (green) in the SNc. The boundary of the SNc was delineated according to the immunofluorescence of TH (red). **e** Normalized density of EGFP-Syb2+ puncta in the SNc of vGlut2-IRES-Cre (vGlut2) and GAD2-IRES-Cre (GAD2) mice as a function of bregma. The normalization was made by dividing the puncta density in the SNc with that in the SC. **f** Example coronal section of ventral midbrain showing the injection of CTB-555 into the SNc of WT mice (left). The boundaries of SNc and VTA were determined by immunofluorescence of TH of dopamine neurons (right). This experiment was repeated independently in five mice with similar results. **g** A schematic diagram showing the distribution of SNc-projecting SC neurons that were labeled by CTB-555. **h, i** Example micrographs (left) and quantitative analyses (right) showing CTB-555+ cells in the SC are predominantly positive for glutamate (**h**) and negative for GABA (**i**). Number of mice was indicated in the graphs (**e, h, i**). Data in **e, h, i** are means ± SEM (error bars). Statistical analyses in **e** were performed by one-way ANOVA (***$P < 0.001$). Scale bars are indicated in the graphs.

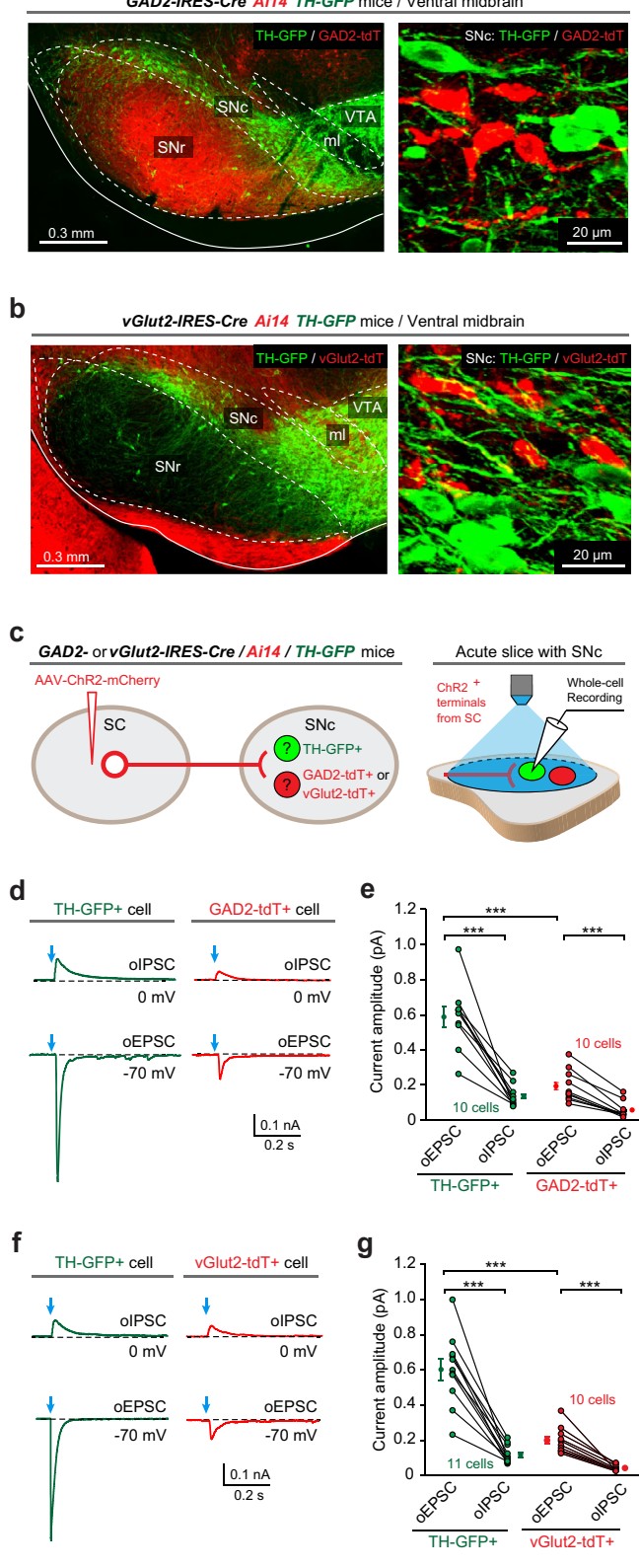

**Fig. 2 Dopamine neurons are the primary postsynaptic target of the SC-SNc pathway. a** An example coronal section with ventral midbrain (left) and the high-magnification micrograph (right) showing the segregation of GAD2-tdT+ neurons and TH-GFP+ neurons in the SNc of GAD2-IRES-Cre/Ai14/TH-GFP triple transgenic mice. This experiment was repeated independently in five mice with similar results. **b** An example coronal section with ventral midbrain (left) and the high-magnification micrograph (right) showing the segregation of vGlut2-tdT + neurons and TH-GFP+ neurons in the SNc of vGlut2-IRES-Cre/Ai14/ TH-GFP triple transgenic mice. This experiment was repeated independently in five mice with similar results. **c** Schematic diagram showing injection of AAV-hSyn-ChR2-mCherry into the SC of GAD2-IRES-Cre/Ai14/TH-GFP or vGlut2-IRES-Cre/Ai14/TH-GFP mice (left) and whole-cell recording from TH-GFP+ (green), GAD2-tdT+ (red), or vGlut2-tdT+ (red) neurons while illuminating ChR2-positive axon terminals from the SC (right). **d, e** Example traces (**d**) and quantitative analyses (**e**) of oEPSCs and oIPSCs from TH-GFP+ and GAD2-tdT+ neurons in the slices of GAD2-IRES-Cre/Ai14/TH-GFP mice. **f, g** Example traces (**f**) and quantitative analyses (**g**) of oEPSCs and oIPSCs from TH-GFP+ and vGlut2-tdT+ neurons in the slices of vGlut2-IRES-Cre/Ai14/ TH-GFP mice. Number of neurons is indicated in the graphs (**e, g**). Data in **e, g** are means ± SEM (error bars). Statistical analyses in **e, g** were performed by one-sided Student's *t*-test (***$P < 0.001$). Scale bars are indicated in the graphs.

mice[44] to label putative SNc vGlut2+ neurons with tdTomato (vGlut2-tdT; Supplementary Fig. 3e, f).

To examine how the SC–SNc pathway synaptically innervates SNc dopamine neurons and GAD2+ neurons, we generated GAD2-IRES-Cre/Ai14/TH-GFP triple transgenic mice. In this mouse line, putative SNc dopamine neurons were genetically labeled by GFP (TH-GFP+), whereas putative GAD2+ neurons were identified as those positive for tdTomato (GAD2-tdT+) (Fig. 2a). AAV-ChR2-mCherry was injected into the SC of GAD2-IRES-Cre/Ai14/TH-GFP mice (Fig. 2c, left). In acute brain slices with the SNc, we illuminated ChR2-mCherry+ axon terminals with light pulses (473 nm, 2 ms) with saturating power (20 mW), while performing whole-cell recordings from TH-GFP+ and adjacent GAD2-tdT+ neurons (Fig. 2c, right). Using low-chloride internal solution[37], we recorded optogenetically evoked excitatory postsynaptic currents (oEPSCs, voltage clamp at −70 mV) and inhibitory postsynaptic currents (oIPSCs, voltage clamp at 0 mV), which were removed by perfusion of antagonists of glutamate receptors (APV and CNQX) and GABAa receptor (picrotoxin), respectively (Supplementary Fig. 3g–i). We found that the amplitude of oEPSCs was significantly higher than that of oIPSCs in both SNc TH-GFP+ and GAD2-tdT+ neurons (Fig. 2d, e). This was consistent with the morphological observation that the SNc-projecting SC neurons are primarily glutamatergic (Fig. 1). Moreover, the amplitude of oEPSCs in TH-GFP+ neurons was significantly higher than that in GAD2-tdT+ neurons (Fig. 2d, e), suggesting that the SC–SNc pathway preferentially innervate SNc dopamine neurons.

To test how the SC–SNc pathway is synaptically connected to SNc vGlut2+ neurons, we generated vGlut2-IRES-Cre/Ai14/TH-GFP triple transgenic mice. In this mouse line, putative SNc dopamine neurons were genetically labeled by GFP (TH-GFP+), whereas putative vGlut2+ neurons were identified as those positive for tdTomato (vGlut2-tdT+) (Fig. 2b). We injected AAV-ChR2-mCherry into the SC of triple transgenic mice (Fig. 2c, left). In acute brain slices with the SNc, we recorded oEPSCs and oIPSCs from TH-GFP+ and adjacent vGlut2-tdT+ neurons (Fig. 2c, right). We found that the amplitude of oEPSCs

SNc dopamine neurons with higher fidelity[41]. We confirmed this observation (Supplementary Fig. 3a, b) and used TH-GFP mice to genetically label SNc dopamine neurons. By crossing Ai14 (ref. [42]) with GAD2-IRES-Cre mice[43], we labeled putative SNc GAD2+ neurons with tdTomato (GAD2-tdT; Supplementary Fig. 3c, d). Similarly, Ai14 was crossed with vGlut2-IRES-Cre

was significantly higher than that of oIPSCs in both TH-GFP+ and vGlut2-tdT+ neurons (Fig. 2f, g). Moreover, the amplitude of oEPSCs in TH-GFP+ neurons was significantly higher than that in vGlut2-tdT+ neurons (Fig. 2f, g). These data indicated that the SC–SNc pathway has a stronger synaptic connection with SNc dopamine neurons than their synaptic connections to GAD2+ or vGlut2+ non-dopamine neurons.

**Activation of the SC–SNc pathway triggers striatal dopamine release**. To further confirm that the SNc dopamine neurons are the primary postsynaptic target of SC–SNc pathway, we examined whether activation of this pathway evoke dopamine release in the dorsal striatum[45]. To monitor dopamine release, we employed genetically encoded GPCR-activation-based dopamine sensor (GRAB-DA sensor) that reports dopamine dynamics of nigrostriatal pathway[46]. AAV-C1V1-mCherry[47] and AAV-GRAB-DA were injected into the SC and dorsal striatum of WT mice (Fig. 3a), followed by implantation of optical fibers above the SNc and dorsal striatum, respectively (Fig. 3b). The viral expression and optical fiber implantation were validated by using immunohistochemistry and slice physiology (Fig. 3c–f). In head-fixed mice walking on a cylindrical treadmill, the fluorescence of GRAB-DA sensor was robustly modulated by spontaneous locomotion (Supplementary Fig. 4a, b). When we aligned the GRAB-DA signals with the initiation of locomotion (Supplementary Fig. 4c), we found that the average GRAB-DA signal started to rise at 125 ± 23 ms before locomotion initiation ($n = 6$ mice; Supplementary Fig. 4d). This control experiment agreed well with recent reports that the activity of SNc dopamine neurons signals locomotion[26,27] and confirmed that the GRAB$_{DA}$ sensor could reliably report dopamine release in the dorsal striatum.

Then we tested whether activation of the SC–SNc pathway triggers dopamine release in the dorsal striatum. In freely moving mice, single light pulses (561 nm, 2 ms, 0–20 mW) stimulating the axon terminals of SNc-projecting SC neurons transiently increased the fluorescence of GRAB-DA sensor in the dorsal striatum (Fig. 3g, h). As a control, no obvious fluorescence changes were observed in striatal neurons expressing EGFP (Fig. 3g, h). Moreover, the light-evoked GRAB-DA signals were abrogated by D2 receptor antagonist haloperidol (Fig. 3i, j). These data indicated that SC–SNc pathway activation triggers dopamine release in the dorsal striatum, supporting that the SNc dopamine neurons are the postsynaptic target of the SC–SNc pathway.

**Single SNc-projecting SC neurons encode locomotion speed**. To test whether the SC–SNc pathway is involved in locomotion, we made single-unit recording from the SNc-projecting SC neurons by using antidromic activation strategy[22]. AAV-ChR2-mCherry[48] was injected into the SC of WT mice, followed by implantation of an optical fiber above the ChR2-mCherry+ axon terminals in the SNc (Supplementary Fig. 5a). Three weeks after viral injection, single-unit recording was performed with a tungsten electrode in the SC of head-fixed awake mice walking on a cylindrical treadmill (Fig. 4a, left). The putative SNc-projecting SC neurons were identified by the antidromic action potentials (APs) evoked by light pulses (473 nm, 1 ms, 2 mW) that illuminated ChR2-mCherry+ axon terminals in the SNc (Fig. 4a, right). The antidromically evoked APs had to conform to two criteria[49,50]. First, their waveform should be similar to that of APs during locomotion. Second, their latencies to light pulses should be less than 5 ms. We obtained 18 units that met the above two empirical criteria. Their antidromically evoked APs possessed waveforms quantitatively correlated with those of APs during locomotion (Fig. 4b and

Supplementary Fig. 5b) and had short response latencies to light pulses (2.7 ± 0.4 ms, $n = 18$ units; Fig. 4c).

Then we examined the instantaneous firing rate of these putative SNc-projecting SC neurons before, during, and after locomotion on the treadmill. In general, all of the units that were antidromically activated (18/18) were modulated by non-predatory treadmill walking (Supplementary Video 1 and Fig. 4d). To examine the temporal relationship between the activities of SNc-projecting SC neurons and locomotion initiation, we aligned firing rate of individual units with the onset of locomotion (Fig. 4e, top). We defined the response onset time as the time when the signal reached 15% of peak amplitude relative to the baseline. The average response curve started to rise at 107 ± 15 ms before locomotion onset ($n = 18$ units; Fig. 4e, bottom). Similarly, we aligned firing rate of individual units with the offset of locomotion (Fig. 4f, top), and found that the average response curve dropped to baseline at 121 ± 16 ms after locomotion offset ($n = 18$ units; Fig. 4f, bottom). To examine how the activities of these units were modulated during locomotion, we plotted the response–speed curve of each single unit[51] and found a correlation between the firing rate and locomotion speed in the range of 3–30 cm/s (Fig. 4g; Spearman correlation coefficient = 0.7528; $P = 7.06E-07$). Histological verification indicated that all the recorded units were localized within the intermediate and deep layers of the SC (Fig. 4h and Supplementary Fig. 5c). These data suggested that the SNc-projecting SC neurons encode locomotion speed of mice. We also analyzed the recorded units that were not antidromically activated ($n = 41$ units), and found they exhibited three types of responses to locomotion initiation (Supplementary Fig. 5d–g).

**The SNc-projecting SC neurons were recruited during appetitive locomotion**. The above single-unit analyses were performed in head-fixed mice and could not reveal how the SC–SNc pathway is recruited in freely moving mice under locomotion-related behavioral contexts. To address this concern, we injected AAV2-retro-DIO-GCaMP7 in the SNc of vGlut2-IRES-Cre mice and implanted an optical fiber above the SNc-projecting SC neurons (Fig. 5a, b). In freely moving mice, GCaMP fluorescence of SNc-projecting SC neurons was recorded before and during appetitive locomotion in various behavioral contexts (e.g., approaching prey, food, or conspecifics). We found that these neurons robustly responded to appetitive locomotion when mice approached prey that was restrained at the corner of arena (Fig. 5c–e). These neurons also responded to appetitive locomotion toward food pellet (Supplementary Fig. 6a–c) and conspecifics (Supplementary Fig. 6d–f). These data suggested that the SNc-projecting SC neurons are recruited during appetitive locomotion.

In response to looming visual stimuli, mice exhibited escape behavior to avoid the imminent threats[52]. The escape behavior was used to measure defensive locomotion[51]. We found that, when the mice escaped from the looming visual stimuli, the GCaMP fluorescence of SNc-projecting SC neurons significantly decreased during defensive locomotion (Fig. 5f–h). These data indicated that the SNc-projecting SC neurons are not recruited by defensive locomotion.

**The SC–SNc pathway is required for appetitive locomotion in predatory hunting**. To explore the role of SC–SNc pathway in locomotion, we synaptically inactivated the SNc-projecting SC neurons with tetanus neurotoxin (TeNT), a molecular tool to block neurotransmitter release by proteolytic cleavage of Syb2 (refs. [53,54]). We employed a dual-AAV strategy[22] by injecting AAV2-retro-mCherry-IRES-Cre[55] and AAV-DIO-EGFP-2A-TeNT[29] into the SNc and SC of WT mice, respectively (Fig. 6a).

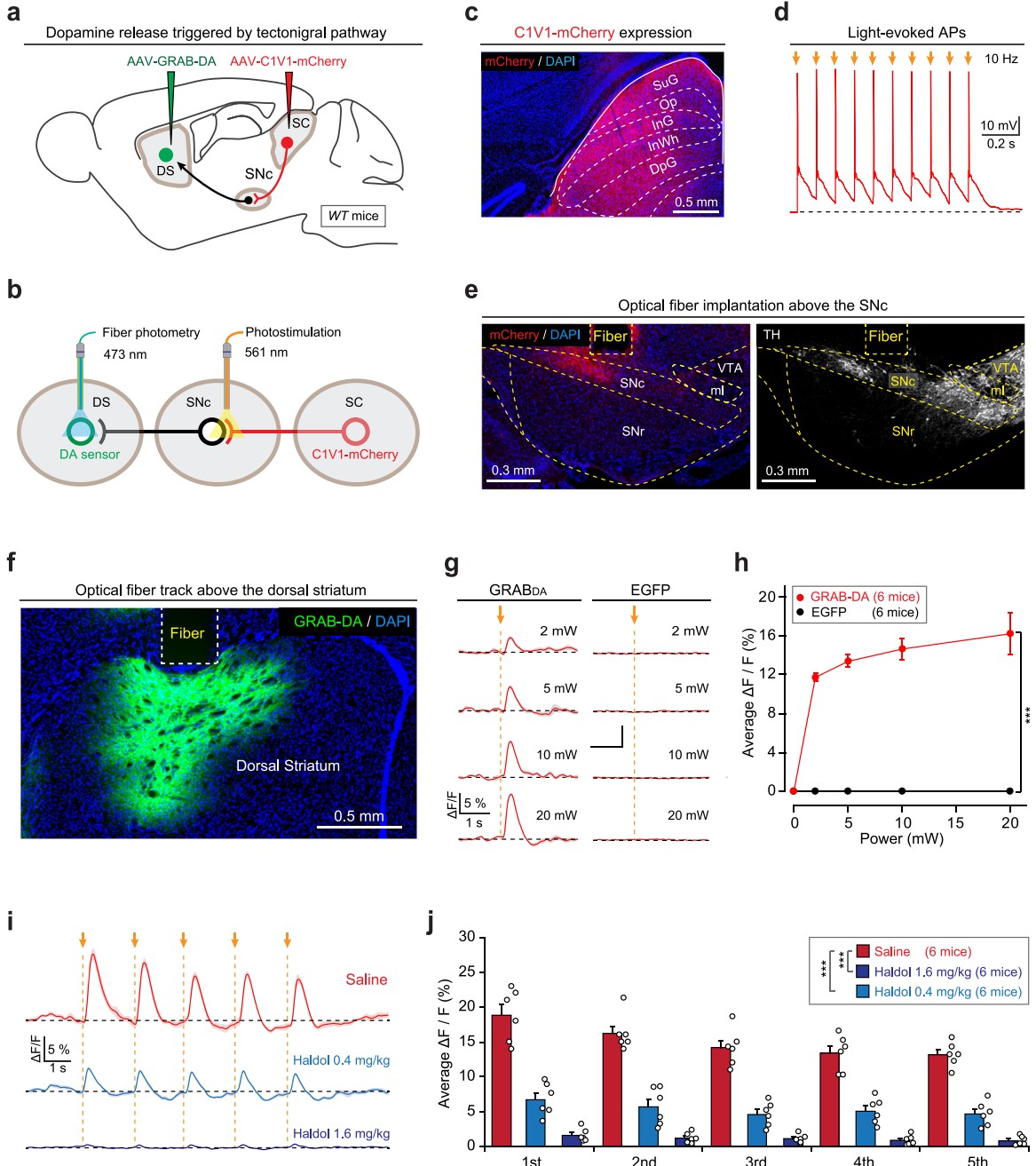

**Fig. 3 Activation of the SC–SNc pathway triggers dopamine release in the dorsal striatum. a** Schematic diagram showing injections of AAV-hSyn-C1V1-mCherry in the SC and AAV-hSyn-GRAB-DA in the dorsal striatum (DS) of WT mice. **b** Schematic diagram showing implantation of optical fibers above the SNc and DS to apply photostimulation and fiber photometry recording, respectively. **c** An example coronal section with C1V1-mCherry expression in the SC. This experiment was repeated independently in five mice with similar results. **d** In acute SC slices, light pulses (2 ms, 561 nm, 10 Hz, 10 pulses) illuminating on C1V1-mCherry+ SC neurons reliably triggered action potential firing. **e** An example coronal section of ventral midbrain showing an optical-fiber track above the C1V1-mCherry+ axon terminals in the SNc (left), the boundary of which was determined according to the immunofluorescence of TH (right). This experiment was repeated independently in six mice with similar results. **f** An example coronal section with an optical-fiber track above the DS neurons expressing GRAB-DA sensor. This experiment was repeated independently in six mice with similar results. **g**, **h** Example traces (**g**) and input–output curve (**h**) of GRAB-DA signals evoked by photostimulation of the SC–SNc pathway with different laser power. EGFP was used as a control of GRAB-DA sensor. **i**, **j** Example traces (**i**) and quantitative analyses (**j**) of GRAB-DA signals evoked by photostimulation (561 nm, 5 pulses, 2 ms, 0.5 Hz, 10 mW) of the SC–SNc pathway in mice treated with saline or haloperidol (0.4 or 1.6 mg/kg). Number of mice was indicated in the graphs (**h**, **j**). Data in **h**, **j** are means ± SEM (error bars). Statistical analyses in **h**, **j** were performed by one-way ANOVA (***P < 0.001). Scale bars are indicated in the graphs.

In a control experiment, we validated the effectiveness of this dual-AAV strategy to specifically silence the SC–SNc pathway without affecting other tectofugal pathways (Supplementary Fig. 7). The injection of AAV2-retro-mCherry-IRES-Cre in the SNc (Supplementary Fig. 8a) and AAV-DIO-EGFP-2A-TeNT in the SC cooperatively labeled SNc-projecting SC neurons with EGFP (Fig. 6b, top), as demonstrated by the co-expression of mCherry and EGFP in the same SC neurons (Fig. 6b, bottom).

Then we analyzed the effects of SC–SNc pathway inactivation on appetitive locomotion in various behavioral contexts. In

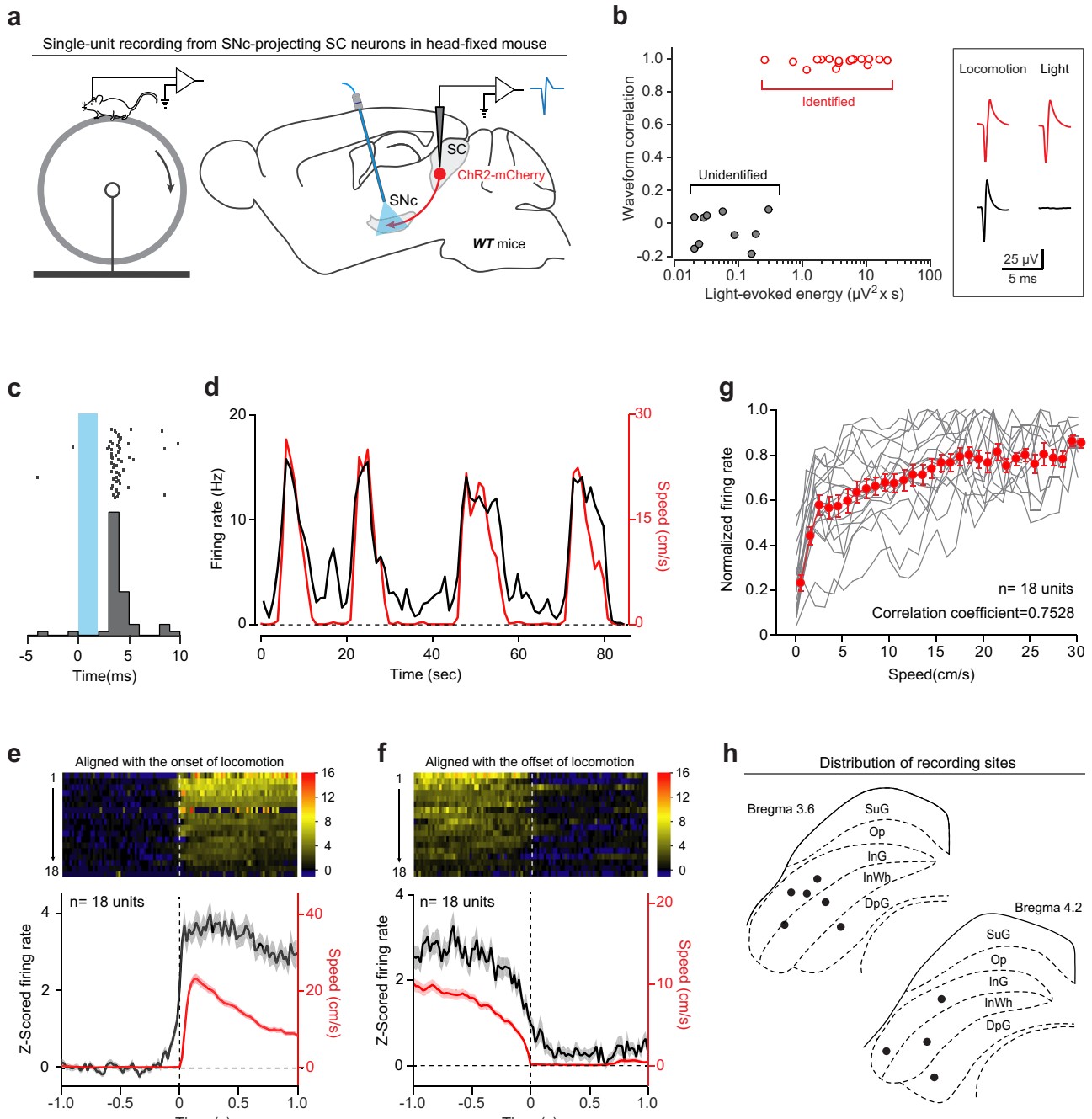

**Fig. 4 SNc-projecting SC neurons encode locomotion speed. a** Schematic diagram showing a head-fixed awake mouse walking on a cylindrical treadmill (left) and antidromic activation strategy for single-unit recording from SNc-projecting SC neurons (right). **b** Correlation analysis of action potentials of individual units evoked either by light pulses (Light) or by locomotion (Locomotion), confirming a segregation between antidromically identified units (red dots and traces) and unidentified units (gray dots and traces). **c** Raster (top) and peristimulus time histogram (PSTH, bottom) of an example putative SNc-projecting SC neurons showing a latency of ~3 ms relative to the onset of light pulses. **d** Smoothed PSTH (trace in black) of an example putative SNc-projecting SC neuron aligned with locomotion speed (trace in red). **e**, **f** Heat-map graphs (top) and averaged time course (bottom) of Z-scored firing rate of 18 putative SNc-projecting SC neurons aligned to the onset (**e**) and the offset (**f**) of locomotion. **g** Averaged (red) and individual (gray) normalized instantaneous firing rate during locomotion as a function of locomotion speed in 500 ms bins. **h** Distribution of recording site (black dots) in the SC. Number of units was indicated in the graphs (**e**–**g**). Data in **e**–**g** are means ± SEM (error bars).

predatory hunting, appetitive locomotion occurred when predator approached prey (Fig. 6c). By measuring the instantaneous azimuth angle and distance between prey and predator (Supplementary Fig. 8b), we were able to identify a series of intermittent approach episodes (Supplementary Fig. 8c, shaded areas in orange) according to the established criteria[3]. Appetitive locomotion in these approach episodes was quantitatively

assessed by measuring the speed of approach and frequency of approach. The speed of approach was calculated by averaging the peak speed of each approach episode in the trial. The frequency of approach was the number of approach episodes divided by total time of the trial. With the method above, we labeled the approach episodes (shaded areas in orange) in the behavioral ethogram of predatory hunting in mice without (Ctrl) and with (TeNT)

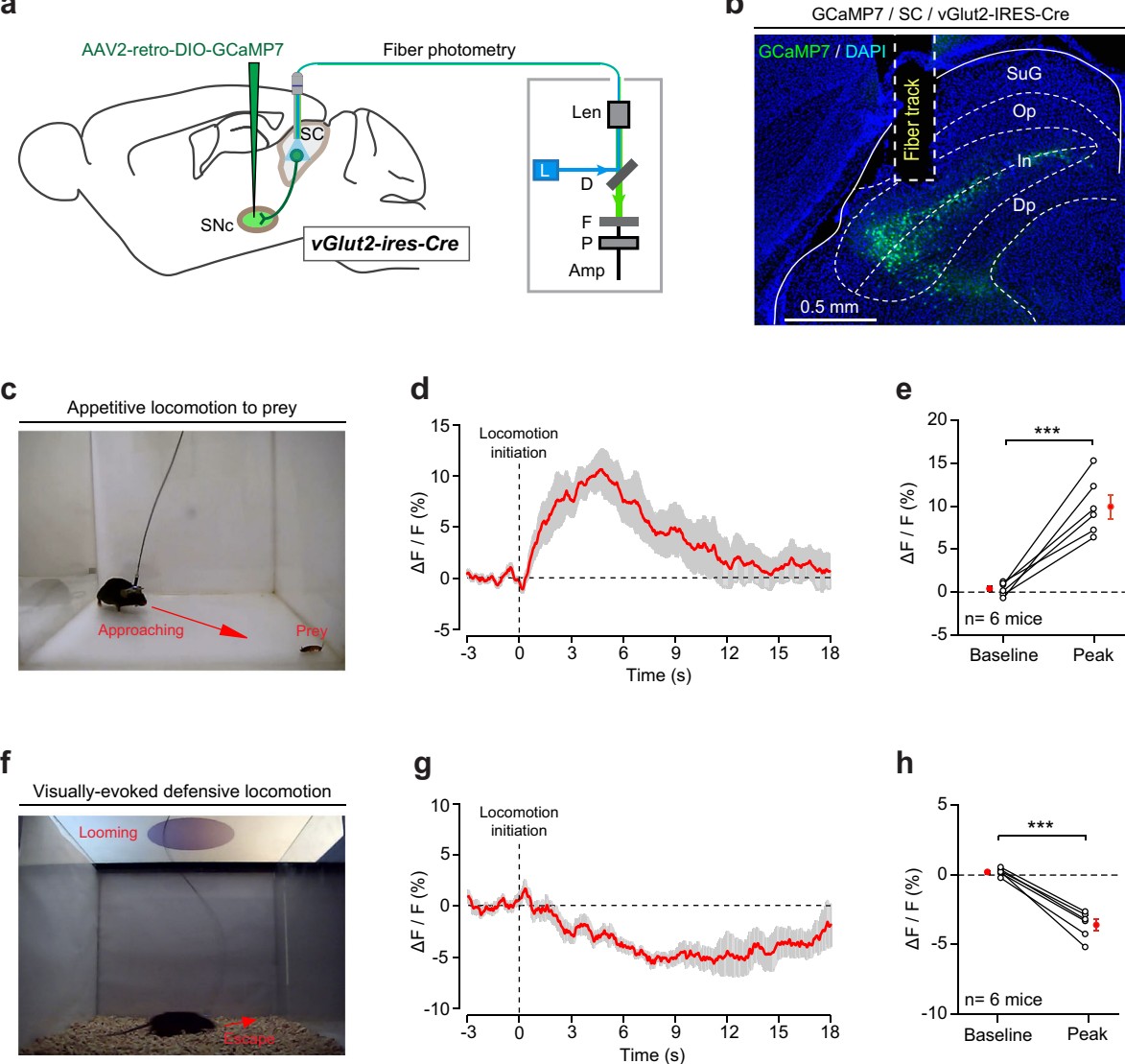

**Fig. 5 SNc-projecting SC neurons are selectively recruited during appetitive locomotion. a** Schematic diagram showing fiber photometry recording from SNc-projecting SC neurons. **b** An example micrograph showing optical-fiber track above the SNc-projecting SC neurons that expressed GCaMP7. This experiment was repeated independently in six mice with similar results. **c**, **f** Example pictures showing fiber photometry recording from SNc-projecting SC neurons when mice exhibit appetitive locomotion toward prey (**c**) or defensive locomotion away from looming visual stimuli (**f**). **d**, **g** Time courses of normalized GCaMP fluorescence of SNc-projecting SC neurons of an example mouse before and during appetitive locomotion toward prey (**d**) or defensive locomotion away from looming visual stimuli (**g**). **e**, **h** Quantitative analyses of GCaMP fluorescence showing the SNc-projecting SC neurons were activated during appetitive locomotion toward prey (**e**) and inactivated during defensive locomotion away from looming visual stimuli (**h**). Scale bars are indicated in the graphs. Number of mice was indicated in the graphs (**e, h**). Data in **d, e, g, h** are means ± SEM (error bars). Statistical analyses in **e, h** were performed by one-sided Student's *t*-test (***$P < 0.001$).

synaptic inactivation of the SC–SNc pathway (Supplementary Videos 2 and 3, Fig. 6d, e and Supplementary Fig. 8d, e). TeNT-mediated inactivation of SC–SNc pathway impaired predatory hunting by significantly increasing the time required for prey capture (Fig. 6f). However, the success rate of predatory hunting, which was ~100% in the 12 mice tested, was not attenuated by synaptic inactivation of the SNc-projecting SC neurons. Such detrimental effect on predatory hunting could not be explained by an impairment of predatory attack, because neither the latency nor the frequency of jaw attack during hunting was changed by SC–SNc pathway inactivation (Fig. 6g, h). In contrast, both the speed of approach and frequency of approach in predatory hunting were significantly decreased (Fig. 6i, j). We found that inactivation of the

SC–SNc pathway also reduced the speed of appetitive locomotion when mice approached food pellet (Supplementary Fig. 8f–h) or conspecifics (Supplementary Fig. 8i–k). These data suggested that the SC–SNc pathway is required for appetitive locomotion in different behavioral contexts.

Next we tested the role of the SC–SNc pathway in defensive and exploratory locomotion. We found that mice without (Ctrl) and with (TeNT) synaptic inactivation of the SC–SNc pathway similarly exhibited escape followed by freezing in response to looming visual stimuli (Supplementary Videos 4 and 5 and Fig. 6k). Quantitative analyses of locomotion speed indicated that synaptic inactivation of the SC–SNc pathway did not alter the peak speed of escape triggered by looming visual stimuli (Fig. 6l)

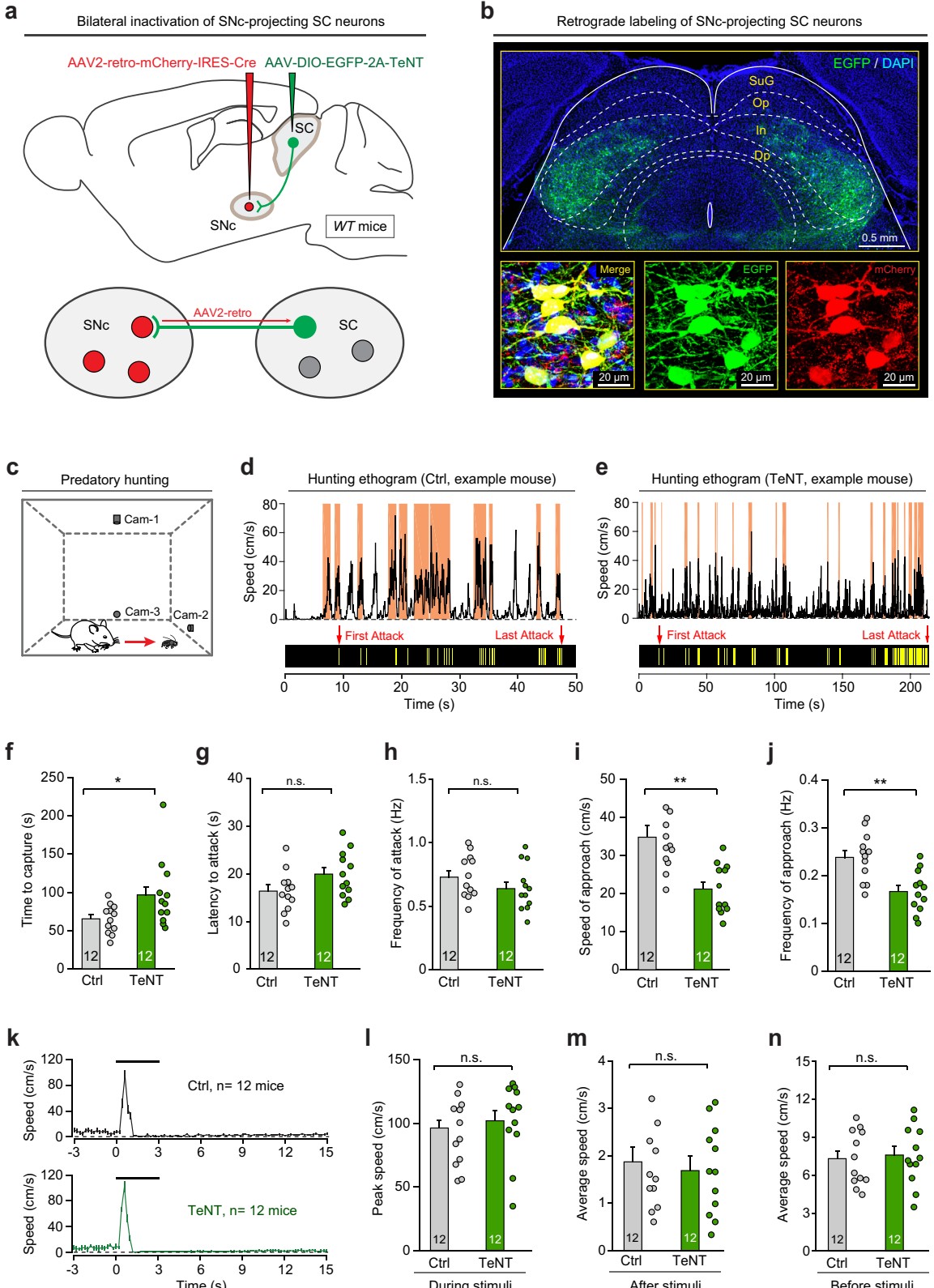

**a** Bilateral inactivation of SNc-projecting SC neurons

AAV2-retro-mCherry-IRES-Cre    AAV-DIO-EGFP-2A-TeNT

**b** Retrograde labeling of SNc-projecting SC neurons

**c** Predatory hunting

**d** Hunting ethogram (Ctrl, example mouse)

**e** Hunting ethogram (TeNT, example mouse)

or average speed after stimuli (Fig. 6m), suggesting this pathway is not required for defensive locomotion. Moreover, these mice exhibited similar average locomotion speed before looming visual stimuli (Fig. 6n), suggesting that the SC–SNc pathway may not be involved in exploratory locomotion as well.

The above loss-of-function data were based on irreversible inactivation of the SNc-projecting SC neurons. To confirm these results, we employed the strategy of designer receptors exclusively activated by designer drugs (DREADD)[56] to chemogenetically silence SNc-projecting SC neurons (Supplementary Fig. 9a–c).

**Fig. 6 SC–SNc pathway is selectively required for appetitive locomotion during predatory hunting. a** Schematic diagram showing the dual-AAV strategy to selectively inactivate the SNc-projecting SC neurons with TeNT. **b** An example coronal brain section showing EGFP+ SNc-projecting SC neurons distributed in the intermediate layer (In) and deep layer (Dp) of SC. Inset, merged and single-channel micrographs showing SNc-projecting SC neurons were dually labeled by mCherry and EGFP. This experiment was repeated independently in 12 mice with similar results. **c** Schematic diagram showing predatory hunting paradigm in an arena. **d**, **e** Behavioral ethograms showing the time courses of locomotion speed (top) and jaw attacks (bottom) during predatory hunting of example mice either without (**d** Ctrl) or with (**e** TeNT) synaptic inactivation of SNc-projecting SC neurons. The shaded areas (orange) indicated the approach episodes in predatory hunting. **f–j** Quantitative analyses of time to capture (**f**), latency to attack (**g**), frequency of attack (**h**), speed of approach (**i**), and frequency of approach (**j**) during predatory hunting in mice without (Ctrl) and with (TeNT) synaptic inactivation of SNc-projecting SC neurons. **k** Time courses of locomotion speed before, during, and after looming visual stimuli in mice without (Ctrl) and with (TeNT) synaptic inactivation of SNc-projecting SC neurons. **l–n** Quantitative analyses of peak locomotion speed during stimuli (**l**), average locomotion speed after stimuli (**m**), and average locomotion speed before stimuli (**n**) of mice without (Ctrl) and with (TeNT) synaptic inactivation of SNc-projecting SC neurons. Number of mice was indicated in the graphs (**f–n**). Data in **f–n** are means ± SEM (error bars). Statistical analyses in **f–j**, **l–n** were performed by one-sided Student's *t*-tests ($^{n.s.}P > 0.1$; *$P < 0.05$; **$P < 0.01$). Scale bars are indicated in the graphs.

We found that chemogenetic inactivation of the SNc-projecting SC neurons by intraperitoneal injection of clozapine N-oxide (CNO, 1 mg/kg) reduced speed of approach and frequency of approach during predatory hunting (Supplementary Fig. 9d–h). However, the defensive locomotion evoked by looming visual stimuli was not altered by chemogenetic inactivation of the SNc-projecting SC neurons (Supplementary Fig. 9i–l). These data further support that the SC–SNc pathway is selectively required for appetitive locomotion.

**The SC–SNc pathway promotes appetitive locomotion in predatory hunting**. To further explore the role of SC–SNc pathway in locomotion, we examined whether activation of this pathway promotes locomotion on a linear runway. AAV-ChR2-mCherry was bilaterally injected into the SC of WT mice (Fig. 7a and Supplementary Fig. 10a), followed by implantation of an optical fiber above the ChR2-mCherry+ axon terminals in the SNc (Fig. 7b). In acute brain slices with the SC, light pulses (10 Hz, 2 ms, 10 mW) reliably evoked action potentials from ChR2-mCherry+ SC neurons (Supplementary Fig. 10b). In a linear runway (Fig. 7c), photostimulation of the SC–SNc pathway (10 Hz, 20 ms, 6 s, 10 mW) increased locomotion speed of mice (Fig. 7d, e and Supplementary Fig. 10c). As a control experiment, we photostimulated the top-down pathway from the primary motor cortex (M1) to the SNc, and found that activation of the M1–SNc pathway did not significantly promote locomotion in mice (Supplementary Fig. 10d–f). These data indicated that activation of the SC–SNc pathway promotes locomotion of mice.

To test how the SC–SNc pathway promotes locomotion, we examined the effects of SC–SNc pathway activation on paw movements when mice walked on the linear runway (Supplementary Fig. 10g). We found that photostimulation of SC–SNc pathway significantly increased both the step frequency and stride length of the fore-paws and hind-paws (Supplementary Fig. 10h, i). These data indicated that activation of the SC–SNc pathway promotes locomotion of mice by increasing the frequency and amplitude of limb movements.

Then we examined whether activation of the SC–SNc pathway boosts appetitive locomotion during predatory hunting (Fig. 7f). We labeled the approach episodes (shaded areas in orange) in the behavioral ethogram of predatory hunting in mice without (OFF) and with (ON) photostimulation of the SC–SNc pathway (10 Hz, 20 ms, 10 mW) (Supplementary Videos 6 and 7; Fig. 7g, h and Supplementary Fig. 10j, k). We found that activation of the SC–SNc pathway significantly increased the speed of approach (Fig. 7i), increased the frequency of approach (Fig. 7j), and reduced the time required for prey capture (Fig. 7k). In contrast, the latency and the frequency of predatory attack with jaw during hunting were not altered by activation of the SC–SNc pathway (Fig. 7l, m). These

data suggested that activation of the SC–SNc pathway promoted appetitive locomotion during predatory hunting.

**The SC–SNc pathway promotes appetitive locomotion via SNc dopamine neurons**. As the SNc dopamine neurons may be the primary postsynaptic target of the SC–SNc pathway (Figs. 2 and 3), we asked whether the SC–SNc pathway promotes appetitive locomotion via SNc dopamine neurons. To address this question, we employed the strategy of DREADD to chemogenetically silence SNc dopamine neurons. AAV-DIO-hM4Di-mCherry was bilaterally injected into the SNc of DAT-IRES-Cre mice[57], resulting in the expression of hM4Di-mCherry primarily in the SNc dopamine neurons (TH+) (Supplementary Fig. 11a). We found that bilateral chemogenetic silencing of these dopamine neurons with intraperitoneal injection of CNO (1 mg/kg) impaired the basal appetitive locomotion during predatory hunting (Supplementary Fig. 11b–d), spontaneous locomotion before looming visual stimuli (Supplementary Fig. 8e), and defensive locomotion during and after looming visual stimuli (Supplementary Fig. 8f, g). These data suggest that the SNc dopamine neurons were required for all three types of locomotion.

To test whether SNc dopamine neurons mediate the appetitive locomotion evoked by SC–SNc pathway activation, we injected AAV-DIO-hM4Di-mCherry and AAV-ChR2-EYFP into the SNc and SC of DAT-IRES-Cre mice bilaterally, followed by implanting two optical fibers above the SNc (Fig. 8a, b). AAV-DIO-mCherry was used as a control of AAV-DIO-hM4Di-mCherry. In the SC, the expression of ChR2-EYFP and the efficiency to evoke action potentials from ChR2-EYFP+ neurons were validated (Supplementary Fig. 11h, i). In the ventral midbrain, hM4Di-mCherry was specifically expressed in SNc dopamine neurons that were intermingled with ChR2-EYFP+ axon terminals from SC neurons (Fig. 8c). Chemogenetic suppression of neuronal firing by CNO (10 μM) was confirmed in slice physiology (Fig. 8d). In mice treated with saline, light stimulation of SC–SNc pathway significantly increased the speed of approach (Fig. 8e, left) and the frequency of approach (Fig. 7g, left) during predatory hunting. When the same mice were intraperitoneally treated with CNO (1 mg/kg) to chemogenetically suppress the activities of SNc dopamine neurons, activation of SC–SNc pathway only mildly increased the speed of approach (Fig. 7e, right) and the frequency of approach (Fig. 7g, right). For each mouse, we calculated "net increase" of approach speed by subtracting speed of approach during laser OFF from that during laser ON. It turned out that chemogenetic inactivation of SNc dopamine neurons with CNO significantly reduced the net increase of approach speed (Fig. 7f). Similarly, we calculated "net increase" of approach frequency by subtracting frequency of approach during laser OFF from that during laser ON. We found that inactivation of SNc dopamine neurons with

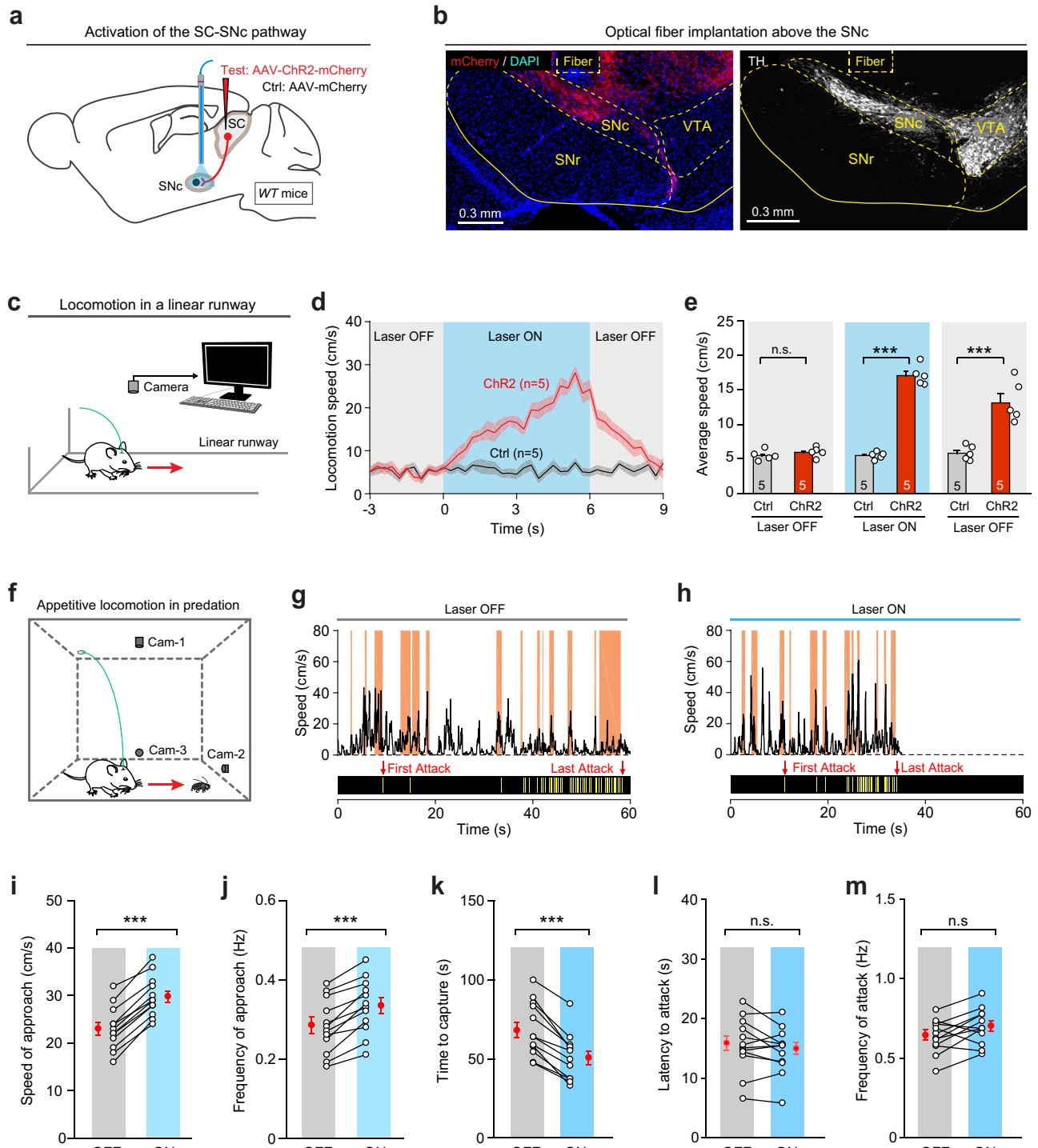

**Fig. 7 Activation of the SC–SNc pathway promoted appetitive locomotion. a** Schematic diagram showing injection of AAV-ChR2-mCherry into the SC of WT mice, followed by optical fiber implantation above the SNc. **b** An example coronal section of ventral midbrain with an optical-fiber track above the ChR2-mCherry+ axon terminals in the SNc (left), the boundary of which was delineated by the immunofluorescence of TH (right). This experiment was repeated independently in 12 mice with similar results. **c** Schematic diagram showing the experimental configuration to monitor mouse locomotor behavior in the linear runway. **d** Time courses of locomotion speed of control (Ctrl) and test mice (ChR2) in the linear runway before, during and after light stimulation of the SC–SNc pathway (10 Hz, 20 ms, 6 s, 10 mW). **e** Quantitative analyses of average locomotion speed of control (Ctrl) and test mice (ChR2) before, during, and after photostimulation of the SC–SNc pathway. **f** Schematic diagram showing the experimental configuration to monitor predatory hunting in the arena. **g, h** Behavioral ethograms showing the time courses of locomotion speed (top) and jaw attacks (bottom) during predatory hunting of an example mouse without (**g**) and with (**h**) photostimulation of the SC–SNc pathway (10 Hz, 20 ms, 10 mW). The shaded areas (orange) indicated the approach episodes in predatory hunting. For the analyses of azimuth angle and PPD, see Supplementary Fig. 10j, k. **i–m** Speed of approach (**i**), frequency of approach (**j**), time to capture (**k**), latency to attack (**l**), and frequency of attack (**m**) in predatory hunting of mice without (OFF) and with (ON) photostimulation of the SC–SNc pathway. Number of mice was indicated in the graphs (**d**, **e**, **i–m**). Data in **d**, **e**, **i–m** are means ± SEM (error bars). Statistical analyses in **e**, **i–m** were performed by one-sided Student's $t$-tests (n.s.$P > 0.1$; ***$P < 0.001$). Scale bars are indicated in the graphs.

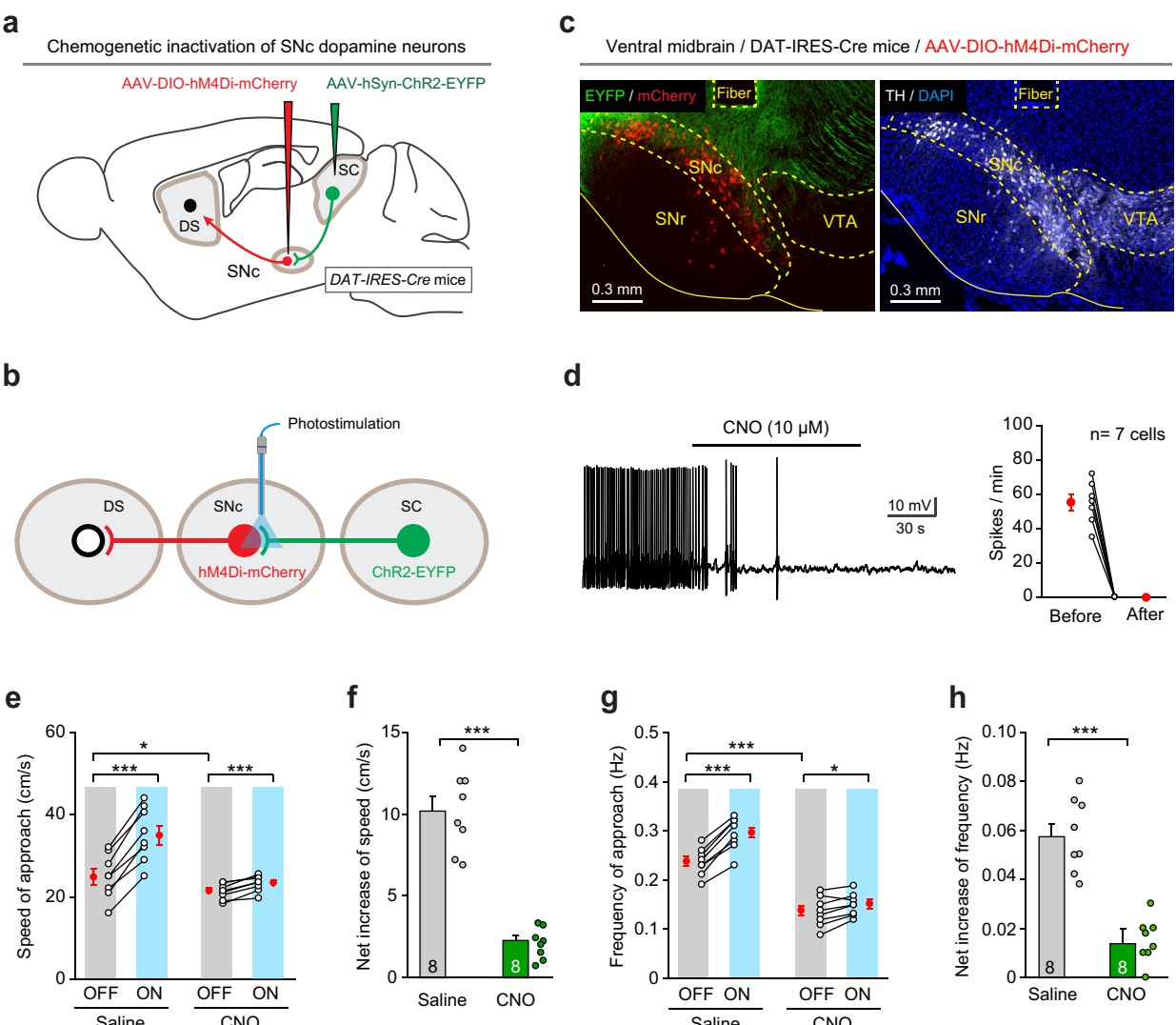

**Fig. 8 SC–SNc pathway promotes appetitive locomotion via SNc dopamine neurons. a** Schematic diagram showing injection of AAV-DIO-hM4Di-mCherry and AAV-ChR2-EYFP into the SNc and SC of DAT-IRES-Cre mice. **b** Schematic diagram showing implantation of an optical fiber above the SNc to apply light stimulation on ChR2-EYFP+ axon terminals from the SC. **c** An example coronal section of ventral midbrain showing the optical fiber track above the hM4Di-mCherry+ SNc dopamine neurons intermingled with ChR2-EYFP+ axons from the SC (left). The boundaries of SNc and VTA were delineated by the immunofluorescence of TH (right). This experiment was repeated independently in eight mice with similar results. **d** An example train of action potentials recorded from SNc dopamine neurons expressing hM4Di-mCherry (left) and quantitative analyses of spike number per minute before and 3 min after perfusion of CNO (10 μM) in ACSF (right). **e** The effect of light stimulation of the SC–SNc pathway on speed of approach during predatory hunting in mice without (Saline) and with (CNO) chemogenetic suppression of SNc dopamine neurons. **f** Chemogenetic suppression of SNc dopamine neurons with CNO significantly attenuated the net effect of SC–SNc pathway activation on the speed of approach. **g** The effect of light stimulation of the SC–SNc pathway on frequency of approach during predatory hunting in mice without (Saline) and with (CNO) chemogenetic suppression of SNc dopamine neurons. **h** Chemogenetic suppression of SNc dopamine neurons with CNO significantly attenuated the net effect of SC–SNc pathway activation on the frequency of approach. Number of cells (**d**) and mice (**e–h**) was indicated in the graphs. Data in **d–h** are means ± SEM (error bars). Statistical analyses in **d–h** were performed by one-sided Student's $t$-tests (*$P < 0.05$; ***$P < 0.001$). Scale bars are indicated in the graphs.

CNO reduced the net increase of approach frequency during predatory hunting (Fig. 7h). Together, these data suggested that the SC–SNc pathway promotes appetitive locomotion via SNc dopamine neurons.

## Discussion

Appetitive locomotion is required for organisms to approach incentive stimuli in goal-directed behaviors. How the brain controls appetitive locomotion is poorly understood. Here we used predatory hunting as a behavioral paradigm to address this question. We demonstrate an excitatory subcortical circuit from the SC to the SNc to boost appetitive locomotion. The SC–SNc pathway transmits locomotion-speed signals to dopamine neurons and triggers dopamine release in the dorsal striatum. Activation of this pathway promoted appetitive locomotion during predatory hunting, whereas synaptic inactivation of this pathway impairs appetitive locomotion rather than defensive locomotion. Together, these data reveal the SC as an important source to provide locomotion-related signals to SNc dopamine neurons to boost appetitive locomotion.

As a naturalistic goal-directed behavior, predatory hunting has been the focus of studies using diverse animal models, such as toad[58], zebrafish[59–61], and rodents[3,4,62]. In these animal models,

it was found that the optic tectum and its mammalian homolog, the SC, play a fundamental role in predatory hunting (toad[58], zebrafish[63,64], rodents[19,20]). In rodents, a recent study has shown that genetically defined neuronal subtypes in the SC make distinct contributions to prey capture behavior in mice[21]. The hunting-associated SC neurons may form divergent neural pathways to orchestrate distinct behavioral actions during predatory hunting, such as attacking prey[22] and, as demonstrated in this study, appetitive locomotion for approaching prey.

In another line of research, it was found that brain areas that were thought to be related to food intake are also involved in predatory hunting. For example, optogenetic activation of GABAergic neurons in the central amygdala (CeA)[4], the lateral hypothalamus (LH)[65], or the ZI[33] provoked strong predatory hunting in mice. The involvement of feeding-related areas in predatory hunting may be evolutionarily conserved, because the inferior lobe of hypothalamus in zebrafish also participates in prey capture behavior[66]. In addition, activation of CaMKIIα-positive neurons in the medial preoptic area, which is related to object craving, also induces hunting-like actions toward prey[67]. Understanding how the neurons in the SC and in these feeding-related brain areas coordinately control predatory hunting is a challenging task for future study.

As an important neuromodulatory system in the brain, dopamine system plays a critical role in conditioned and unconditioned appetitive behaviors[68,69]. Earlier studies using systemic treatment of agonists or antagonists of dopamine receptors have demonstrated strong effects on predatory hunting in mammals[70–72]. However, two critical questions remained unanswered. First, how is dopamine system recruited during predatory hunting? Second, considering the multiple clusters of dopamine neurons in the brain, which specific clusters of dopamine neurons participate in modulating predatory hunting? In this study, we show that the dopamine neurons in the SNc are innervated by the SC, a central hub to orchestrate predatory hunting. The SC–SNc pathway may provide locomotion-related signals to SNc dopamine neurons to boost appetitive locomotion during predatory hunting. These results may provide some clues to the above unanswered questions. They also supported the recent studies showing the involvement of SNc dopamine neurons in the vigor of body movements[24–28].

In their seminal studies, Redgrave and colleagues[73] proposed that the SC–SNc pathway may serve as a route for salient visual stimuli to drive phasic activities of dopamine neurons. In primate, this pathway may mediate visually evoked reward expectation signals in dopamine neurons during reinforcement learning[74]. In the present study, we recorded single-unit activity of SNc-projecting SC neurons in head-fixed walking mice (Supplementary Video 1), and unexpectedly found that the SNc-projecting SC neurons encode locomotion speed (Fig. 4). This observation prompted us to examine the role of SC–SNc pathway in regulating locomotion during predatory hunting. Our data may have added another perspective for understanding the functions of the SC–SNc pathway. Although we did not systematically examine the sensory responses of the recorded neurons, we do not rule out the possibility that these neurons may respond to salient sensory stimuli (e.g., visual or vibrissal tactile stimuli). In future study, it will be interesting to explore whether the SC–SNc pathway can integrate both sensory and locomotion-related signals to dynamically modulate appetitive locomotion during hunting.

In the present study, an interesting observation is that the SNc-projecting SC neurons were activated during appetitive locomotion but suppressed during defensive locomotion. Such observation supports the specific role of the SC–SNc pathway in appetitive locomotion. But we have not explored the neural mechanisms underlying this phenomenon. According to a recent work by Kiehn's group[51], the gait and speed of locomotion are encoded by distinct neuronal populations in the mesencephalic locomotion region. For example, vGlut2+ neurons in the pedunculopontine nucleus encode speed in a slower range that were used during exploratory and appetitive locomotion. In contrast, vGlut2+ CnF neurons encode locomotion with higher speed, which is usually used during escape. We found that SNc-projecting SC neurons encode locomotion speed in a slower range (0–30 cm/s). It is likely that the speed of defensive locomotion may be encoded by other SC neurons, such as CnF-projecting SC neurons, a hypothesis remains to be tested in future study.

Where do the locomotion-speed signals of the SNc-projecting SC neurons originate? Several motor-related brain areas (e.g., SNr, PPTg, and motor cortex) directly project to the SC and may provide motor signals to the SC[75]. This speculation was supported by a recent study showing that the projection from the SNr is the strongest among the above motor-related brain areas[76]. The axons of GABAergic SNr neurons terminate in the lateral part of deep layers of the SC[77], a region that contains SNc-projecting SC neurons studied here. The inhibition and excitation of SNr neurons well predict the initiation and suppression of locomotion, respectively[78]. These studies suggested that locomotion-related signals of SNc-projecting SC neurons may at least partially originate from the SNr, which is the primary output of basal ganglia.

## Methods

**Animals**. All experimental procedures were conducted following protocols approved by the Administrative Panel on Laboratory Animal Care at the National Institute of Biological Sciences, Beijing (NIBS). The vGlut2-IRES-Cre[44], GAD2-IRES-Cre[43], DAT-IRES-Cre[57], TH-GFP[40], and Ai14[42] mouse lines were imported from the Jackson Laboratory (JAX Mice and Services). Mice were maintained on a circadian 12-h light/12-h dark cycle with food and water available ad libitum. Mice were housed in groups (3–5 animals per cage) before they were separated 3 days prior to virus injection. After injection of viral vectors, each mouse was housed in one cage for 3 weeks before subsequent experiments. To avoid potential sex-specific differences, we used male mice only.

**AAV vectors**. We used two AAV serotypes (AAV-DJ, AAV2-retro) in this study. The AAVs used in this study are listed in Supplementary Table 1. The viral particles were purchased from Shanghai Taitool Bioscience Inc. and Brain VTA Inc. The titers of viral vectors were initially in the range of $0.8–1.5 \times 10^{13}$ particles/ml. The final titer used for AAV injection after dilution with PBS is $5 \times 10^{12}$ viral particles/ml.

**Stereotaxic injection**. We anesthetized mice with intraperitoneal injection of tribromoethanol (125–250 mg/kg). Mice received standard surgery that exposed the brain surface above the SC, substantia nigra pars compacta (SNc), VTA, ZI, lateral posterior thalamic nucleus (LPTN), primary motor cortex (M1), or dorsal striatum. Coordinates used for SC injection were bregma −3.60 mm, lateral ±1.30 mm, and dura −1.75 mm. Coordinates used for SNc injection were bregma −3.40 mm, lateral ±1.25 mm, and dura −4.00 mm. Coordinates used for VTA injection were bregma −3.40 mm, lateral ±0.50 mm, and dura −4.00 mm. Coordinates used for ZI injection were bregma −2.00 mm, lateral ±1.25 mm, and dura −4.25 mm. Coordinates used for LPTN injection were bregma −2.18 mm, lateral ±1.35 mm, and dura −2.25 mm. Coordinates used for M1 injection were bregma 1.18 mm, lateral ±1.75 mm, and dura −0.50 mm. Coordinates used for dorsal striatum injection were bregma 0.74 mm, lateral ±1.50 mm, and dura −2.40 mm. The AAVs and CTB were injected with a glass pipette that was connected to a Nano-liter Injector 201 (World Precision Instruments, Inc.) with a flow rate as slow as 0.15 μl/min to ensure local brain tissue was not damaged. The injection pipette stayed at the injection site for at least 20 min after injection was completed and then was slowly withdrawn from the brain.

For optogenetic activation and synaptic inactivation experiments, AAV injections were bilateral. For anterograde and retrograde tracing experiments, CTB injection was unilateral. Histological analyses were conducted 1 week (for CTB) and at least 3 weeks (for AAV) after injection. Experimental designs related to viral injection are summarized in Supplementary Table 2.

**Optical-fiber implantation**. After the pipette was withdrawn from the brain, a ceramic ferrule with an optical fiber (230 μm in diameter, numerical aperture = 0.37) was implanted with the fiber tip on top of the SNc (bregma −3.40 mm, lateral ±1.25 mm, and dura −3.80 mm), the dorsal striatum (bregma 0.74 mm, lateral

±1.50 mm, and dura −2.20 mm), or the SC (bregma 3.80 mm, lateral ±1.75 mm, and dura −1.75 mm). After implantation, the ferrule was secured on the skull with dental cement. The surgical wound was treated with antibiotics and then the skin was sutured. The experiments of optogenetics and fiber photometry were conducted around 3 weeks after optical fiber implantation. All experimental designs related to optical fiber implantation are summarized in Supplementary Table 2. For optogenetic stimulation, the output of the laser was measured and adjusted to 2, 5, 10, 15, and 20 mW before each experiment. The pulse onset, duration, and frequency of light stimulation were controlled by a programmable pulse generator attached to the laser system.

**Single-unit recording**. An electrophysiological strategy of antidromic activation was used to identify the single-unit of SNc-projecting SC neurons. AAV-hSyn-ChR2-mCherry was injected into the SC of wild-type mice, followed by an optical fiber implanted above the SNc. Three weeks after viral injection, single-unit recording was performed with a tungsten electrode that was vertically inserted into the lateral SC of head-fixed awake mouse. The electrode was advanced into the SC with a Narishige micro-manipulator. The spikes were amplified by a differential amplifier (Model 1800, A-M Systems, Everett, WA, USA), digitized (10 kHz), and recorded with Spike2 software (Version 7.03). When the single-unit activities were isolated, we tested whether they were from SNc-projecting SC neurons. The putative SNc-projecting SC neurons were identified by the antidromic spikes evoked by light pulses (473 nm, 1 ms, 2 mW) that illuminated ChR2-mCherry+ axon terminals in the SNc. The antidromically evoked spikes were expected to conform to two criteria. First, their latency to the pulses of light illumination should be less than 5 ms. Second, their waveform should be quantitatively comparable to that of spikes evoked by locomotion (Fig. 4b). Only units with action potentials faithfully following the light stimulations with latency shorter than 5 ms were used to test for locomotion-correlated activity (Fig. 4c). We performed spike sorting with Spike2 Software. For a certain train of action potentials, after setting the threshold, Spike2 automatically generated the templates and performed the spike sorting. The quality of spike clustering was further confirmed by principal component analysis (Supplementary Fig. 5b). We recorded the single-unit activities of SNc-projecting SC units, while simultaneously measuring the instantaneous locomotion speed of mice walking on the treadmill (Nanjing Thinktech Inc.).

**Verification of recording sites**. We marked the recording sites of the putative SNc-projecting SC neurons with electrolytic lesions applied by passing positive currents (40 μA, 10 s) through the tungsten electrode. Under deep anesthesia with urethane, the brain was perfused with saline and phosphate-buffered saline (PBS) containing 4% paraformaldehyde (PFA). After regular histological procedure, frozen sections were made at 40 μm in thickness and counterstained with DAPI for histological verification of recording sites.

**Preparation of behavioral tests**. After AAV injection and fiber implantation, the mice were housed individually for 3 weeks before the behavioral tests. They were handled daily by the experimenters for at least 3 days before the behavioral tests. On the day of behavioral test, we moved the mouse cages to the testing room and habituated the mice to the room conditions for 3 h before the experiments started. The apparatus was cleaned with ethanol (20%) to eliminate odor cues from other mice. The behavioral tests were performed in the same circadian period (1 PM–7 PM). All behaviors were scored by the experimenters who were blind to the animal treatments.

**Behavioral paradigm for predatory hunting**. The procedure of predatory hunting experiment followed a published work[22]. Before the predatory hunting test, the mice went along a habituation procedure that lasted 9 days (Days H1–H9). On each of the first three habituation days (Days H1, H2, and H3), three cockroaches were placed in the home cage with no food restriction in mice at 2 PM. The mice readily captured and consumed the prey within 3 h after prey introduction. On Day H3, H5, H7, and H9, we initiated 24-h food deprivation at 7 PM by removing chow from the home cages. On Day H4, H6, and H8 at 5 PM, we let the mice freely explore the arena (25 cm × 25 cm) for 10 min, followed by three practice trials of predatory hunting for cockroach. After hunting practice, we put the mice back in their home cages and returned the chow at 7 PM. On the test day, we let the mice freely explore the arena for 10 min, followed by the introduction of a cockroach. For each mouse, predatory hunting was repeated for three trials. Each trial began with the introduction of prey to the arena. The trial ended when the predator finished ingesting the captured prey. After the mice finished ingesting the prey body, debris was removed before the new trial began. After the tests, the mice were placed back in their home cages, followed by the return of chow. The cockroach was purchased from a merchant in Tao-Bao Online Stores (www.taobao.com).

**Measurement of appetitive locomotion and predatory attack in predatory hunting**. In the paradigm of predatory hunting, mouse behavior was recorded in the arena with three orthogonally positioned cameras (50 frames/s; Point Grey Research, Canada). With the video taken by the overhead camera, the instantaneous head orientation of predator relative to prey (azimuth angle) and predator–prey distance (PPD) was analyzed with the Software EthoVision XT 14

(Noldus Information Technology). The episode of approach was identified by two empirical criteria[3]. First, the PPD should continuously decrease until it is less than 3 cm. Second, the azimuth angle of mouse head to cockroach should be within the range of −90° to +90°. In WT mice, each trial of predatory hunting contains 10–20 episodes of approach. Speed of approach and frequency of approach were used to quantitatively measure the appetitive locomotion in the episodes of approach. Speed of approach of each mouse was calculated by averaging the peak speed in all the approach episodes in the trial. Frequency of approach was the total number of approach episodes divided by the time to prey capture in the trial.

With the videos taken by the two horizontal cameras, we carefully identified predatory attacks with jaw by replaying the video frame by frame (50 frames/s). We marked the predatory jaw attacks with yellow vertical lines in the behavioral ethogram of predatory hunting. With this method, we measured three parameters of predatory hunting, such as time to prey capture, latency to jaw attack, and frequency of jaw attack. Time to prey capture was defined as the time between the introduction of prey and the last jaw attack. Latency to jaw attack was defined as the time between the introduction of the prey and the first jaw attack from the predator. Frequency of jaw attack was defined as the number of jaw attacks divided by time to prey capture. Data for three trials were averaged.

**Measurement of defensive locomotion triggered by looming visual stimuli**. Measurement of defensive locomotion triggered by looming visual stimulus followed a published work[35]. Defensive locomotion was measured in an arena (35 cm × 35 cm, square open field) with corn-cob bedding. No shelter was used. A regular computer monitor was positioned above the arena to present overhead looming visual stimuli. After entering, the mice explored the arena for 10 min. Then three cycles of overhead looming visual stimuli consisting of an expanding dark disk were presented. The visual angel of the dark disk was expanded from 2° to 20° within 250 ms. Luminance of the background and the dark disk were 3.6 and 0.1 cd/m², respectively. Mouse behaviors were recorded by two orthogonally positioned cameras (50 frames/s; Point Grey Research, Canada) with infrared illumination provided by LEDs. The instantaneous location of the mouse in the arena was measured by a custom-written Matlab program, according to a published work[35]. The instantaneous locomotion speed was calculated with a 200 ms time-bin. The Matlab code is available upon request.

**Measurement of locomotion in linear runway**. Mouse behavior was recorded in the linear runway (10 cm × 16 cm × 120 cm) with an overhead camera (50 frames/s; Point Grey Research, Canada). With the video taken by the overhead camera, we measured the instantaneous locomotion speed with the Software EthoVision XT 14 (Noldus Information Technology).

**Measurement of paw movements in linear runway**. Mouse behavior was recorded in the linear runway (10 cm×16 cm × 120 cm) with a camera (50 frames/s; Point Grey Research, Canada) placed beneath the transparent floor of the runway. With the video taken by the camera, an in-house Matlab program was used to plot the position of the four paws as the mouse walked in the runway. The stride length and step frequency were automatically measured with the Matlab program offline. The Matlab code is available upon request.

**Slice physiological recording**. Preparation of acute brain slices was performed according to the published work[79]. The brains of adult mice anesthetized with isoflurane were rapidly removed and placed in ice-cold oxygenated (95% O₂ and 5% CO₂) cutting solution (228 mM sucrose, 11 mM glucose, 26 mM NaHCO₃, 1 mM NaH₂PO₄, 2.5 mM KCl, 7 mM MgSO₄, and 0.5 mM CaCl₂). Coronal brain slices (400 μm) were prepared with a vibratome (VT 1200 S, Leica Microsystems, Wetzlar, Germany). The slices were incubated at 28 °C in oxygenated artificial cerebrospinal fluid (ACSF 125 mM NaCl, 2.5 mM KCl, 1.25 mM NaH₂PO₄, 1.0 mM MgCl₂, 25 mM NaHCO₃, 15 mM glucose, and 2.0 mM CaCl₂) for 30 min (~305 mOsm, pH 7.4). The slices were then kept at room temperature under the same conditions for 30 min before moving to the recording chamber at room temperature. The ACSF was perfused at 1 ml/min. The acute brain slices were visualized with a ×40 Olympus water immersion lens, differential interference contrast optics (Olympus Inc., Japan), and a CCD camera.

Patch pipettes were made by pulling the borosilicate glass capillary tubes (Cat #64-0793, Warner Instruments, Hamden, CT, USA) using a PC-10 pipette puller (Narishige Inc., Tokyo, Japan). For recording of postsynaptic currents in voltage-clamp mode, pipettes were filled with solution (126 mM Cs-methanesulfonate, 10 mM HEPES, 1 mM EGTA, 2 mM QX-314 chloride, 0.1 mM CaCl₂, 4 mM Mg-ATP, 0.3 mM Na-GTP, 8 mM Na-phosphocreatine, pH 7.3 adjusted with CsOH, ~290 mOsm), according to a published work[37]. For recording of action potentials in current-clamp mode, pipettes were filled with solution (135 mM K-methanesulfonate, 10 mM HEPES, 1 mM EGTA, 1 mM Na-GTP, 4 mM Mg-ATP, pH 7.4). The resistance of pipettes varied between 3.0 and 3.5 MΩ. The current and voltage signals were recorded with MultiClamp 700B and Clampex 10 data acquisition software (Molecular Devices). After establishment of the whole-cell configuration and equilibration of the intracellular pipette solution with the cytoplasm, series resistance was compensated to 10–15 MΩ. Recordings with series resistances higher than 15 MΩ were rejected. We used an optical fiber (230 μm in

diameter, numerical aperture = 0.37) to deliver light pulses, with the fiber tip positioned 500 μm above the acute brain slices. Laser power was adjusted to 2, 5, 10, or 20 mW. Light-evoked action potentials from ChR2-mCherry+ neurons in the SC were triggered by a light-pulse train (473 nm, 2 ms, 10 Hz, 20 mW) synchronized with Clampex 10 data acquisition software (Molecular Devices). Optically evoked postsynaptic currents from SNc neurons were triggered by single light pulses (2 ms) in the presence of 4-aminopyridine (4-AP, 20 μM) and tetrodotoxin (TTX, 1 μM). To examine the neurotransmitter/receptor type of optically evoked postsynaptic currents, we perfused D-AP5 (50 μM)/CNQX (20 μM) or picrotoxin (PTX, 50 μM) within ACSF. Data were analyzed with Clampfit 10 data analysis software (Molecular Devices).

**Fiber photometry system**. A fiber photometry system (ThinkerTech, Nanjing, China) was used to record either GRAB_DA signals from genetically identified neurons[46] or the SNc-projecting SC neurons that expressed GCaMP7. To induce fluorescence signals, a laser beam from a laser tube (488 nm) was reflected by a dichroic mirror, focused by a ×10 lens (NA 0.3) and coupled to an optical commutator. A 2-m optical fiber (230 μm in diameter, NA 0.37) guided the light between the commutator and implanted optical fiber. To minimize photo bleaching, the power intensity at the fiber tip was adjusted to 0.02 mW. The fluorescence of GRAB-DA or GCaMP was band-pass filtered (MF525-39, Thorlabs) and collected by a photomultiplier tube (R3896, Hamamatsu). An amplifier (C7319, Hamamatsu) was used to convert the photomultiplier tube current output to voltage signals, which were further filtered through a low-pass filter (40 Hz cut-off; Brownlee 440). The analog voltage signals digitalized at 100 Hz were recorded by a Power 1401 digitizer and Spike2 software (CED, Cambridge, UK).

**Fiber photometry recording of GRAB_DA signals**. AAV-GRAB-DA was stereo-taxically injected into the dorsal striatum of WT mice followed by optical fiber implantation above the injected site (see "Stereotaxic injection" and "Optical-fiber implantation"). Two weeks after AAV injection, fiber photometry was used to record GRAB-DA signals from the cell bodies of dorsal striatum neurons in freely moving mice. A flashing LED triggered by a 1-s square-wave pulse was simultaneously recorded to synchronize the video and GRAB-DA signals. For recordings from freely moving mice, mice with optical fibers connected to the fiber photometry system freely explored the arena for 10 min. After the experiments, the optical fiber tip sites above the dorsal striatum neurons were histologically examined in each mouse.

**Fiber photometry recording of SNc-projecting SC neurons**. AAV2-retro-DIO-GCaMP7 was stereotaxically injected into the SNc of vGlut2-IRES-Cre mice followed by optical fiber implantation above the intermediate layer of the lateral SC (see "Stereotaxic injection" and "Optical fiber implantation"). Three weeks after AAV injection, fiber photometry was used to record GCaMP fluorescence from the cell bodies of SNc-projecting SC neurons in freely moving mice. A flashing LED triggered by a 1-s square-wave pulse was simultaneously recorded to synchronize the video and GCaMP signals. The GCaMP fluorescence was recorded before, during, and after appetitive and defensive locomotion in various behavioral contexts. For measuring appetitive locomotion toward prey and food pellet, mice were food-deprived for 18 h before the experiment. For measuring appetitive locomotion toward conspecifics, a female mouse was used. After the experiments, the optical fiber tip sites above the SC were histologically examined in each mouse.

**Histological procedures**. Mice anesthetized with isoflurane were sequentially perfused with saline and PBS containing 4% PFA. Then the brains were removed and incubated in PBS containing 30% sucrose until they sank to the bottom. Post-fixation of the brain was avoided to optimize immunohistochemistry of GABA and glutamate. Cryostat sections (40 μm) containing the SC, SNc, LPTN, ZI, or dorsal striatum were collected, incubated overnight with blocking solution (PBS containing 10% goat serum and 0.7% Triton X-100), and then treated with primary antibodies diluted with blocking solution for 3–4 h at room temperature. Primary antibodies used for immunohistochemistry are listed in Supplementary Table 1 and also listed here: Rabbit Polyclonal Anti-EGFP (Abcam, CAT# ab290; Lot# GR3196305-l; dilution 1:2000), Rabbit Polyclonal Anti-mCherry (Abcam, CAT# ab167453; Lot# GR587698; dilution 1:2000), Rabbit Polyclonal Anti-Glutamate (Sigma, CAT# G6642; Lot# 116H4815; dilution 1:500), Rabbit Polyclonal Anti-GABA (Sigma, CAT# A2052; Lot# 238K2568; dilution 1:500), and Rabbit Polyclonal Anti-TH (Sigma, CAT# AB152; dilution 1:500). Primary antibodies were washed three times with washing buffer (PBS containing 0.7% Triton X-100) before incubation with secondary antibodies (tagged with Cy2, Cy3, or Cy5; dilution 1:500; Life Technologies Inc., USA) for 1 h at room temperature. The secondary antibodies are listed here: Goat anti-Rabbit Alexa Fluor 488 (A11034), Goat Anti-Rabbit Alexa Fluor 546 (A11010), Goat anti-mouse Alexa Fluor 488 (A11001), and Goat Anti-Rabbit Alexa Fluor 546 (A11030). Sections were then washed three times with washing buffer, stained with DAPI, and washed with PBS, transferred onto Super Frost slides, and mounted under glass coverslips with mounting media.

Sections were imaged with an Olympus (Japan) VS120 epifluorescence microscope (×10 objective lens) or an Olympus FV1200 laser scanning confocal microscope (×20 and ×60 oil-immersion objective lens). Samples were excited by 488, 543, or 633 nm lasers in sequential acquisition mode to avoid signal leakage. Saturation was avoided by monitoring pixel intensity with Hi-Lo mode. Confocal images were analyzed with ImageJ software.

**Quantification of synaptic puncta density**. The micrographs used for measuring puncta density of SynaptoTag (Fig. 1b, d) were acquired with a ×63 objective of Zeiss confocal system and analyzed with NIH ImageJ. The analysis of the synaptic puncta followed a published work[80]. In brief, the scale of micrographs was set in NIH ImageJ based on the physical dimension of micrographs acquired by Zeiss confocal system. After converting the micrographs from RGB color mode to 16-bit mode, the puncta in micrographs were binarized and the puncta density was measured automatically by NIH ImageJ. Then the puncta density in the SNc of each mouse was normalized by dividing with that in the intermediate layer of the lateral SC (Fig. 1e).

**Cell-counting strategies**. Cell-counting strategies are summarized in Supplementary Table 3. For counting cells in the SC, we collected 40-μm coronal sections from bregma −3.28 to bregma −4.48 for each mouse. Six sections evenly spaced by 200 μm were sampled for immunohistochemistry to label cells positive for different markers. We acquired micrographs (×10 objective, Olympus FV1200 microscope, Japan) within intermediate and deep layers of the SC followed by cell counting with ImageJ software. We calculated the percentages of glutamate+ and GABA+ neurons in the neuronal population retrogradely labeled by CTB-555. For counting cells in the SNc, we collected coronal sections (40 μm) from bregma −2.80 to bregma −3.80 for each mouse. Five sections evenly spaced by 200 μm were sampled for immunohistochemistry to label SNc cells positive for different markers. After image acquisition, we outlined the SC and SNc followed by cell counting with ImageJ software. The boundary of SNc in coronal sections was identified based on TH staining.

**Data quantification and statistical analysis**. All experiments were performed with anonymized samples in which the experimenter was unaware of the experimental conditions of the mice. For the statistical analyses of experimental data, Student's t-test and one-way ANOVA were performed with Origin Professional 6.0. The "n" used for these analyses represents number of mice or cells. See the detailed information of statistical analyses in figure legend and in Supplementary Table 4.

**Reporting summary**. Further information on research design is available in the Nature Research Reporting Summary linked to this article.

## Data availability
All data supporting the findings of this study are provided within the paper and its supplementary information. All additional information will be made available upon reasonable request to the authors. Source data are provided with this paper.

## Code availability
The MATLAB code for data analyses is available from the corresponding author upon request.

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

## Acknowledgements

We thank Drs. Thomas Südhof, Karl Deisseroth, and Minmin Luo for providing plasmids and mouse lines. This work was supported by the National Natural Science Foundation of China (31925019 and 31671095 to P.C., 31771150 to Y.W., Top talent program of Hebei province to F.Z.), and the open funds of the State Key Laboratory of Medical Neurobiology. All data are archived in NIBS.

## Author contributions

P.C., C.S., D.L., J.Z., M. He., Y.L., F.Z., Y.W., and X.Q. conceived the study. C.S., M. Huang, Z.X., X.C., Q.P., and A.L. did injections and fiber implantation. C.S., Z.C., and H.G. did slice physiology. C.S. M. Huang, and D.L. did single-unit recording and analyses. X.C., C.S., and Q.P. did fiber photometry recording and analyses. C.S., X.C., Z.X., Q.P., H.G., and Y.X. did behavioral tests. D.L., C.S., X.C., Z.X., H.G., Z.C., and P.C. analyzed behavioral data. M. Huang, Q.P., and X.Z. did histological analyses. F.S. did the construction of plasmids for AAV. P.C. wrote the manuscript.

## Competing interests

The authors declare no competing interests.
