## [Peer Review File · Nature Communications]

Reviewers' Comments:

Reviewer #1:

Remarks to the Author:

In this study, Huang et. al. identified a role of SC-SNc pathway in appetitive locomotion. They found that the pathway can bi-directionally modulate the locomotion speed during predation. They further demonstrated that SC preferentially project onto the dopaminergic cells in the SNc and can cause dopamine release in the dorsal striatum. Blocking SNc DA cell activity occlude the locomotion promoting effect of SC-SNc activation. Overall, the study is well preformed and the authors convincingly demonstrated a role of SC-SNc in locomotion. However, whether the pathway plays a specific role in appetitive locomotion requires additional experiments.

Here are some specific comments:

1. Figure S1 and S2 demonstrated that SC cells that project to SNc and VTA or SNc and ZI are largely distinct. Please calculate the probability of overlap by chance and compare it with the observed overlap to understand whether preferentially non-overlapping SC cells project to these regions.
2. Do SC-SNc cells differ from untagged SC cells in any way? Do the SC cells that do not project to the SNc also encode locomotion information?
3. SC-SNc cells increase activity during locomotion in a headfixed preparation, suggesting that the SC-SNc cells do not fire in a context specific manner. However, functional manipulation found that SC-SNc inactivation impairs only locomotion during predation not defense. How to reconcile the recording and functional results? If SC-SNc cells are specifically relevant for appetitive locomotion instead of locomotion per se, the SC-SNc cells are expected to respond differently during different types of locomotion, e.g. defensive locomotion vs. appetitive locomotion. In vivo recording of SC-SNc cells during different types of locomotion in freely moving animals will be essential for addressing whether the role of SC-SNc cells in locomotion is context specific or not.
4. Does SC-SNc pathway play a role in other appetitive locomotion (approach)? How about approaching towards a food pellet or another mouse?
5. When the SC-SNc cells are activated and inactivated, it appears that the structure of the locomotion becomes different. For example, In Figure 3H, the locomotion bouts appear to be shorter. Is this also the case if the animal is headfixed on a running wheel? This may suggest that the pathway is important for controlling certain features of locomotion. The authors are encouraged to analyze the functional and recording data in more details to elucidate the specific locomotion information encoded by the cells.

Reviewer #2:

Remarks to the Author:

The paper by Huang et al focuses on the SC projections to SNc underlying appetitive locomotion, approaching food and/or hunting. They use a wide range of methods to show the circuit properties, anatomy, and functional role of the SC-SNc pathway. The paper is mostly well-written and figures are clear. Some of the experiments should be described in more detail and there are issues that should be addressed before publication.

1. TeNT vs chemo or optogenetics: in order to show that the SC-SNc pathway is not affecting other types of locomotion such as escape from looming stimuli, the authors permanently block the synaptic output of SC-SNc neurons and show no difference in the looming response but changes in the hunting approach locomotion. The synaptic inactivation of SC-SNc cells is done by a dual AAV approach but there is no control showing that the synapses are inactivated. How efficient is the TeNT inactivation of the pathway? This could be done by injecting cre-dependent chr2 together with the cre dependent TeNT and test the pathway in slice. Also, what other targets of the SC cells are affected? Do the same inactivated cells excite other regions that could affect other behaviors in addition to the approach locomotion? The authors should show other YFP expressing regions following the dual-AAV injections. The TeNT inactivation is irreversible and should be confirmed by optogenetic or chemogenetic inactivation of SC-SNc terminals using the same dual-AAV approach. This could show the effects on both approach behavior and looming response in a reversible manner in the same animals.

2. The supplementary movies could be improved by including more information on what is shown instead of requiring going back and forth to the text for context. Information such as the light on/off periods, which are not clear in the video, the frequency of stimulation, the type of mouse and manipulation (TeNT, control, WT, AAV, etc...) would be helpful. Also, if the main parameter is the duration until the last attack, it would be helpful to add the time count explicitly. Perhaps even showing the control and manipulated mice examples side-by-side in the video.
3. The results shown suggest that there was always a successful and purposeful hunting of the cockroaches by the mice. Was this always the case? Were there cases where there was no interest or no success in the hunting? If so, such cases should be mentioned, especially if they were more prevalent in the SC-SNc pathway inactivation experiments.
4. The criteria for choosing the 18 cells in figure 2 are not completely clear. Were there 18 cells that responded with short latency to antidromic light-stim? Were all of them found to be active spontaneously during non-predatory treadmill walking? If not, how many cells were antidromically activated and excluded from analysis (did not show activity during spontaneous locomotion)? This point should be more explicitly described in the text and figure.
5. The experiments depicted in Figure 7 are not selective: bilateral chemogenetic silencing of dopamine SNC neurons is not specific, and is expected to cause impairment of movement in general. Was there any change in locomotion following CNO administration? Was there any change in "hunting" behavior? Spontaneous movement? Escape in looming? The authors show the net effect of ChR2 activation of SC-SNc terminals on such behaviors in the presence and absence of CNO but it is hard to understand how the CNO by itself affected the behavior, especially when there was unspecific silencing of SNc dopaminergic neurons (likely with some spillover to VTA).

Minor:

1. In figure 7G the units seem different from 7H (net change in frequency of approaches).
2. The MS should be checked again for grammar, a few examples:
 - Line 45: "predator employs" should be "predators employ"
 - Line 46: "How the brain control appetitive" should be "controls"
 - Line 57: "of the SC enable itself to orchestrate" should be "enable it to..."
 - Line 126: "we made single-unit recording from" should be "recordings..."

Reviewer #3:

Remarks to the Author:

This manuscript by Huang and colleagues investigated the role of SC-SNc pathway in predatory hunting related motor control. In particular, the authors used a combination of anterior and retrograde circuit tracing, optogenetic and chemogenetic activation/silencing of specific pathway, in vivo and ex vivo electrophysiology and behavioral analysis and convincingly showed quite a few interesting findings: First, a specific group of glutamatergic but not GAD positive SC neurons project to SNc. Second, activation of these SC neurons can excite SNc neurons, and evoke DA release in the dorsal striatum. Third, the SC-SNc pathway is particularly important for predatory hunting-related motor behavior. Importantly, the authors showed that inactivation of these projection (by using specific expression of TeNT in SC-SNc projection neurons) only impaired appetitive locomotion, but not defensive (using looming stimulation) or exploratory locomotion. This set of experiments provided a comprehensive understanding of a specific excitatory projection (SC) onto SNc, and how this projection controls a specific motor behavior.

The authors use anatomical analysis to determine input from the superior colliculus (SC) to the SNc, and identified these inputs from the SC to be prominent for glutamatergic (vGlut2) neurons, and much sparser for GAD2+ neurons. Using retrograde tracing, they describe the existence of different SC neurons projecting to VTA, ZI and SNc. Even though the authors used non-specific method (CTB), not DA neuron specific (for example, Rabies virus approach), the combination of anterior, retrograde and slice physiology approaches alleviated such concern. The in vivo electrophysiology recordings are convincing, and the analysis is thorough. Recent studies have shown that dopamine neurons receive convergent synaptic inputs from broad brain regions. The current work sets up a great example for teasing out how different synaptic inputs contribute to the locomotion control in different behavioral contexts. The current study used technologically innovative approaches, the results are clear and nicely presented. The major conclusion is

supported by the experimental results. As dopamine neurons have been shown to be tightly related to locomotion control and its importance in neurological diseases such as Parkinson's disease, this work would be of interest to a broad neuroscience audience. Overall, the current study is timely, and highly significant. There are no major concerns regarding the experimental design, data interpretation or conclusion. However, I do have a few suggestions below, which can further improve the manuscript.

1) To better determine the functional role of SC-SNc pathway in appetitive locomotion, the authors need to compare the SC-SNc pathway with other excitatory afferents to the SNc, for example, the M1-SNc pathway.

2) The authors showed that activation of SC-SNc pathway could evoke dopamine release in the dorsal striatum, as evidence by pulsed GRAB-DA signals that were suppressed by Haloperidol in a dose-dependent manner. Although these data are strong, it would be more convincing if the author can provide more physiological data. For example, are these GRAB-DA signals modulated by locomotion, on a treadmill or during appetitive locomotion of predatory hunting?

3) Regarding the specificity of TH-GFP mice to label SNc DA neurons (Fig. S7), the authors should perform better analysis. For example, brain sections containing the SNc at different bregma from anterior to posterior should all be included in the analysis. Such detailed analyses should also be done for GAD2-IRES-Cre/Ai14 mice and vGlut2-IRES-Cre/Ai14 mice.

4) The sequence of the presentation. The session on SC-SNc pathway innervates SNc DA neuron (slice physiology) and evoked dopamine release should be introduced before predatory locomotion. This way, the whole story starts from anatomy, to physiology, and end with in vivo and behavior. However, this is reviewer's suggestion. This does not affect the conclusion of the current study.

Authors' response to the reviewers' comments for Huang et al., " **The SC-SNc pathway boosts appetitive locomotion in predatory hunting**", and changes made in the revised manuscript

We very much appreciate the reviewers' careful evaluation of our manuscript and their positive and constructive comments. We are now submitting the revised manuscript to fully address all the reviewers' concerns. Many of the reviewers' comments were very helpful, and we have performed a series of additional experiments to address these comments. Specifically, we have:

1. Made more anatomical analyses to examine how the SC-SNc pathway is segregated from other tectofugal pathways (Supplementary Fig. 2e, 2f, & 2k).
2. Analyzed the responses to treadmill locomotion of single units that were not antidromically activated by light pulses (Supplementary Fig. 5d-5g).
3. Recorded SNc-projecting SC neurons with fiber photometry in freely moving mice during appetitive or defensive locomotion in various behavioral contexts (Fig. 5 and Supplementary Fig. 6).
4. Made more behavioral analyses to examine whether the SC-SNc pathway is required for other types of appetitive locomotion when mice approach conspecifics or food pellet (Supplementary Fig. 8, f-k).
5. Made more behavioral analyses to examine whether the SC-SNc pathway regulates the step frequency and stride length of paw movements during locomotion on a linear runway (Supplementary Fig. 10, g-i).
6. Performed control experiments to validate the efficiency and specificity of TeNT to suppress neurotransmitter release from the SNc-projecting SC neurons (Supplementary Fig. 7).
7. Employed chemogenetic inactivation strategy to confirm that the SNc-projecting SC neurons are required for appetitive locomotion during predatory hunting (Supplementary Fig. 9).
8. Confirmed that chemogenetic inactivation of SNc dopamine neurons induced non-specific suppression of various types of locomotion (Supplementary Fig. 11a-11g).
9. Examined whether activation of the M1-SNc pathway increased locomotion speed of mice, in order to better determine the functional role of SC-SNc pathway in appetitive locomotion (Supplementary Fig. 10, d-f).
10. Examined whether dopamine release occurs during treadmill locomotion (Supplementary Fig. 4).
11. Performed a more detailed quantitative analyses to examine the colocalization of TH and fluorescent proteins (TH-GFP, GAD2-tdT, vGlut2-tdT) in the SNc as a function of bregma (Supplementary Fig. 3a-3f).

We hope that with these additions and the changes introduced into the revised manuscript, it can now be accepted for publication. In the following, we cite the reviewers' comments in full in *italic* typeface, and then provide our answers in **bold** type face. For the convenience of the reviewers, we have appended the related figures and legends behind the answers.

Reviewer #1 :

In this study, Huang et al. identified a role of SC-SNc pathway in appetitive locomotion. They found that the pathway can bi-directionally modulate the locomotion speed during predation. They further demonstrated that SC preferentially project onto the dopaminergic cells in the SNc and can cause dopamine release in the dorsal striatum. Blocking SNc DA cell activity occludes the locomotion promoting effect of SC-SNc activation. Overall, the study is well preformed and the authors convincingly demonstrated a role of SC-SNc in locomotion. However, whether the pathway plays a specific role in appetitive locomotion requires additional experiments.

We thank the reviewer for these positive comments and careful evaluation of our manuscript. We have performed a series of new experiments to test whether the SC-SNc pathway plays a specific role in appetitive locomotion. The new data, which are presented below, support that this pathway plays a specific role in appetitive locomotion.

Here are some specific comments:

1. Figure S1 and S2 demonstrated that SC cells that project to SNc and VTA or SNc and ZI are largely distinct. Please calculate the probability of overlap by chance and compare it with the observed overlap to understand whether preferentially non-overlapping SC cells project to these regions.

We thank the reviewer for this great suggestion. To better evaluate the anatomical relationship between the SC-SNc pathway and other tectofugal pathways, it is essential to calculate the probability of overlap by chance and compare it with the observed overlap. In the revised manuscript, we have measured the overlap of SC-SNc and SC-LPTN pathway, the latter of which is involved in visually-evoked innate fear response. As these two pathways are known to be highly segregated (Wei et al., 2015; Shang et al., 2018), their overlap may reflect the probability of overlap by chance. We found that injection of CTB-488 and CTB-555 into the SNc and LPTN resulted in very few dually-labeled SC neurons (Supplementary Fig. 2e and 2f). Quantitative analyses indicated that only 6.2% SNc-projecting SC neurons innervate the LPTN, whereas only 5.2% LPTN-projecting SC neurons innervate the SNc (Supplementary Fig. 2k).

With similar anatomical analyses on the overlap of the SC-SNc and SC-VTA pathways, we found that 11.3% SNc-projecting SC neurons project to the VTA, whereas 9.7% VTA-projecting SC neurons project to the SNc. Regarding the SC-SNc and SC-ZI pathways, 9.8% SNc-projecting SC neurons project to the ZI, whereas 11.1 % ZI-projecting SC neurons project to the SNc. These data indicate that only a small proportion of SNc-projecting SC neurons send collaterals to the VTA and ZI. These data suggest that the SC-SNc pathway is largely segregated from the SC-VTA, SC-ZI and SC-LPTN pathways in mice. The text related to these changes have been highlighted in the revised manuscript (Line 104 - 129).

Supplementary Fig. 2 The SC-SNc pathway is anatomically segregated from other tectofugal pathways. (a) Example coronal section of the ventral midbrain showing injection of CTB-488 and CTB-555 into the SNc and VTA (left), the boundaries of which were delineated according to the immunofluorescence of TH (right). (b) An example coronal section of the SC (left) and the corresponding illustration (right) showing the distribution of CTB-488+ & CTB-555+ cells in the SC. (c) Example coronal brain sections showing injection of CTB-488 and CTB-555 into the SNc (left) and ZI (right), respectively. (d) An example coronal section of the SC (left) and the corresponding illustration (right) showing the distribution of CTB-488+ & CTB-555+ cells in the SC. (e) Example coronal brain sections showing injection of CTB-488 and CTB-555 into the SNc (left) and LPTN (right), respectively. (f) An example coronal section of the SC (left) and the corresponding illustration (right) showing the distribution of CTB-488+ & CTB-555+ cells in the SC. (g) Example coronal section of the ventral

midbrain showing injection of mixed CTB-488 & CTB-555 into the SNc (*left*), the boundary of which was delineated according to the immunofluorescence of TH (*right*). (**h**) An example coronal section of the SC (*left*) and the corresponding illustration (*right*) showing the distribution of CTB-488+ & CTB-555+ cells in the SC. (**i-l**) Quantitative analyses of the number of cells labeled by CTB-488 and/or CTB-555 in the anterior SC (from Bregma -3.08 mm to -3.80 mm) and posterior SC (from Bregma -3.80 mm to -4.60 mm), showing how the SC-SNc pathway anatomically relates to the SC-VTA pathway (i), the SC-ZI pathway (j), the SC-LPTN pathway (k), and the SC-SNc pathway itself (as a control, l). Scale bars are labeled in the graphs. Numbers of mice (i-l) are indicated in the graphs. Data in (i-l) are means \pm SEM.

2. Do SC-SNc cells differ from untagged SC cells in any way? Do the SC cells that do not project to the SNc also encode locomotion information?

To address this concern, we went through our original data and found that the untagged SC cells (41 units), which were not antidromically activated by light pulses, can be grouped into three classes according to their response to locomotion initiation (Supplementary Fig. 5, d-g). Class I units did not change firing rate to locomotion initiation (Supplementary Fig. 5d). Class II units were inhibited when locomotion was initiated (Supplementary Fig. 5e). Class III units increased firing rate at the initiation of locomotion (Supplementary Fig. 5f). The text related to these changes have been highlighted in the revised manuscript (Line 236 - 238).

Supplementary Fig. 5. (d-f) Three example untagged units that were not antidromically activated, showing unchanged (d), decreased (e) and increased (f) firing rate (black trace) when locomotion was initiated (red trace). (g) Summary of the 41 untagged units that were not identified as SNc-projecting SC neurons, showing various types of relationship between firing rate and locomotion initiation. Data in (d-f) are means \pm SEM (error bars).

3. SC-SNc cells increase activity during locomotion in a head-fixed preparation, suggesting that the SC-SNc cells do not fire in a context specific manner. However, functional manipulation found that SC-SNc inactivation impairs only locomotion during predation not defense. How to reconcile the recording and functional results? If SC-SNc cells are specifically relevant for appetitive locomotion instead of locomotion per se, the SC-SNc cells are expected to respond differently during different types of locomotion, e.g. defensive locomotion vs. appetitive locomotion. In vivo recording of SC-SNc cells during different types of locomotion in freely moving animals will be essential for addressing whether the role of SC-SNc cells in locomotion is context specific or not.

We agree with the reviewer that our data in the original manuscript were not sufficient to show that SNc-projecting SC neurons specifically encode appetitive locomotion. To address this concern, we performed new experiments by recording SNc-projecting SC neurons with fiber photometry in freely moving mice (Fig.5, a and b). We found that these neurons were robustly recruited during appetitive locomotion when mice approached prey (Fig.5, c-e). In contrast, the GCaMP fluorescence of these neurons were decreased during defensive locomotion when the mice escaped from the looming visual stimuli (fig.5, f-h). These new data strongly suggest that the SNc-projecting SC neurons specifically encode appetitive locomotion rather than defensive locomotion. The text related to these changes have been highlighted in the revised manuscript (Line 239 - 248; Line 253 - 258).

Fig. 5 SNc-projecting SC neurons are activated during appetitive locomotion. (a) Schematic diagram showing fiber photometry recording from SNc-projecting SC neurons. (b) An example micrograph showing optical-fiber track above the SNc-projecting SC neurons that expressed GCaMP7. (c, f) Example pictures showing fiber photometry recording from SNc-projecting SC neurons when mice exhibit appetitive locomotion toward prey (c) or defensive locomotion away from looming visual stimuli (f). (d, g) Time courses of normalized GCaMP fluorescence of example mice showing the activity of SNc-projecting SC neurons before and during appetitive locomotion toward prey (d) or during defensive locomotion from looming stimuli (g). (e, h) Quantitative analyses of GCaMP fluorescence showing the SNc-projecting SC neurons were activated during appetitive locomotion toward prey (e) and inactivated during defensive locomotion away from looming visual stimuli (h). Scale bars are indicated in the graphs. Number of mice was indicated in the graphs (e, h). Data in (d, e, g, h) are means \pm SEM (error bars). Statistical analyses in (e, h) were performed by Student t-test (***) $P < 0.001$. For the P values, see Table S4.

To further clarify the specific role of the SC-SNc pathway in appetitive locomotion, we made discussion in the revised manuscript (Line 444-456). According to a recent work by Kiehn's group (Caggiano et al., 2018), the gait and speed of locomotion are in fact encoded by distinct neuronal populations in the mesencephalic locomotion region (LMR). For example, vGlut2+ neurons in the pedunculopontine nucleus encode speed in a slower range that may be used during exploratory and appetitive locomotion. In contrast, vGlut2+ CnF neurons encode locomotion with higher speed, which is likely used during escape. We found that SNc-projecting SC neurons encode locomotion speed in a slow range. It is likely that the speed of defensive locomotion (peak speed > 80 cm/s) may be encoded by other SC neurons, such as SC-CnF pathway, a hypothesis remains to be tested in future study.

4. Does SC-SNc pathway play a role in other appetitive locomotion (approach)? How about approaching towards a food pellet or another mouse?

To answer this question, we performed a new experiment by recording SNc-projecting SC neurons with fiber photometry in freely moving mice. We found that the SNc-projecting SC neurons were also recruited during appetitive locomotion toward food pellet and conspecifics (Supplementary Fig. 6). The text related to these changes have been highlighted in the revised manuscript (Line 249 - 252).

Supplementary Fig. 6 SNc-projecting SC neurons are activated during appetitive locomotion to food pellet and conspecifics. (a, d) Example pictures showing fiber photometry recording from SNc-projecting SC neurons when mice exhibit appetitive locomotion toward food pellet (a) or conspecifics (d). (b, e) Time courses of normalized GCaMP fluorescence of example mice showing the activity of SNc-projecting SC neurons before and during appetitive locomotion toward food pellet (b) or conspecifics (e). (c, f) Quantitative analyses of GCaMP fluorescence showing the SNc-projecting SC neurons were activated during appetitive locomotion toward food pellet (c) and conspecifics (f). Numbers of mice are indicated in the graphs (c, f). Data in (b, c, e, f) are means \pm SEM (error bars).

To further test whether the SC-SNc pathway play a role in other appetitive locomotion, we tested the effect of SC-SNc pathway inactivation on locomotion speed when mice approached food pellet or conspecifics. We found that inactivation of SNc-projecting SC neurons with TeNT reduced the locomotion speed when the mice approached food pellet or conspecifics (Supplementary Fig.8, f-k). Together, these data suggest that the SC-SNc pathway is also required for other forms of appetitive locomotion. The text related to these changes have been highlighted in the revised manuscript (Line 292 - 296).

Supplementary Fig.8f-8k Inactivation of SNc-projecting SC neurons reduced locomotion speed when mice approached food and conspecifics. (f, g) Time courses of locomotion speed of example mice before and during appetitive locomotion toward food pellet without (Ctrl, f) and with (TeNT, g) synaptic inactivation of SNc-projecting SC neurons. (h) Quantitative analyses of average locomotion speed during appetitive locomotion toward food pellet in mice without (Ctrl) and with (TeNT) synaptic inactivation of SNc-projecting SC neurons. (i, j) Time courses of locomotion speed of an example mouse before and during appetitive locomotion toward conspecifics without (Ctrl, i) and with (TeNT, j) synaptic inactivation of SNc-projecting SC neurons. (k) Quantitative analyses of average locomotion speed during appetitive locomotion toward conspecifics in mice without (Ctrl) and with (TeNT) synaptic inactivation of SNc-projecting SC neurons. Numbers of mice are indicated in the graphs (h, k). Data in (f-k) are means \pm SEM. Statistical analyses (h, k) are performed using Student t-test (** $P < 0.01$).

5. When the SC-SNc cells are activated and inactivated, it appears that the structure of the locomotion becomes different. For example, In Figure 3H, the locomotion bouts appear to be shorter. Is this also the case if the animal is head-fixed on a running wheel? This may

suggest that the pathway is important for controlling certain features of locomotion. The authors are encouraged to analyze the functional and recording data in more details to elucidate the specific locomotion information encoded by the cells.

We thank the reviewer for this great suggestion. To address this concern, we performed a new experiment to test the effects of SC-SNc pathway activation on paw movements when mice walked on the linear runway (Supplementary Fig. 10g). We found that photostimulation of SC-SNc pathway significantly increased both the step frequency and stride length of the fore-paws and hind-paws (Supplementary Fig. 10h and 10i). These data indicated that activation of the SC-SNc pathway promotes locomotion of mice by increasing the frequency and amplitude of limb movements. The text related to these changes have been highlighted in the revised manuscript (Line 331 - 337).

Supplementary Fig. 10g-10i Activation of SNc-projecting SC neurons increased step frequency and stride length. (g) Schematic diagram showing the experimental configuration to monitor paw movements of mice walking on the linear runway. (h, i) Quantitative analyses of step frequency (h) and stride length (i) of fore-paws (*left*) or hind-paws (*right*) in mice walking on the linear runway without (OFF) and with (ON) activation of the SC-SNc pathway (10 Hz, 10 ms, 10 mW).

Reviewer #2 :

The paper by Huang et al focuses on the SC projections to SNc underlying appetitive locomotion, approaching food and/or hunting. They use a wide range of methods to show the circuit properties, anatomy, and functional role of the SC-SNc pathway. The paper is mostly well-written and figures are clear. Some of the experiments should be described in more detail and there are issues that should be addressed before publication.

We thank the reviewer for these positive comments and careful evaluation of our manuscript. Please see our new experimental data below to address the issues in the comments.

1. TeNT vs. chemo- or optogenetics: in order to show that the SC-SNc pathway is not affecting other types of locomotion such as escape from looming stimuli, the authors permanently block the synaptic output of SC-SNc neurons and show no difference in the looming response but changes in the hunting approach locomotion. The synaptic inactivation of SC-SNc cells is done by a dual AAV approach but there is no control showing that the synapses are inactivated. How efficient is the TeNT inactivation of the pathway? This could be done by injecting Cre-dependent ChR2 together with the Cre-dependent TeNT and test the pathway in slice. Also, how are other targets of the SC cells affected? Do the same inactivated cells excite other regions that could affect other behaviors in addition to the approach locomotion? The authors should show other YFP expressing regions following the dual-AAV injections.

We agree with the reviewer that it is essential to determine the efficiency and specificity of TeNT to inactivate the SC-SNc pathway. To address this concern, we performed the control experiment as suggested by the reviewer (Supplementary Fig. 7). AAV2-retro-Cre was injected into the SNc of WT mice, followed by injection of mixture of AAV-DIO-EGFP-2A-TeNT and AAV-DIO-ChR2-mCherry (1:1) into the ipsilateral SC (Supplementary Fig. 7a). We used AAV-DIO-EGFP as a control of AAV-DIO-EGFP-2A-TeNT. This strategy resulted in coexpression of EGFP-2A-TeNT and ChR2-mCherry in the same SNc-projecting SC neurons (Supplementary Fig. 7b and 7c). In the SNc, we found axons co-labeled by both EGFP and mCherry (Supplementary Fig. 7d, left). In contrast, there were sparse axons in other target regions of the SC, such as the VTA (Supplementary Fig. 7d, middle) and the ZI (Supplementary Fig. 7d, right). These new data provide anatomical evidence to support the specificity of the dual-AAV strategy to selectively label SNc-projecting SC neurons.

In acute brain slices, we illuminated ChR2-mCherry+ axons with saturating light (20 mW) and recorded light-evoked postsynaptic currents (PSCs) at -70 mV from neurons in the SNc, VTA and ZI (Supplementary Fig. 7e). We found that light-evoked PSCs in SNc neurons were almost completely abolished by the expression of TeNT (Ctrl: 96 ± 15 pA, n= 8 cells; TeNT: 9.5 ± 1.1 pA, n= 8 cells; Supplementary Fig. 7f), thus validating the efficiency of TeNT to inactivate the SNc-projecting SC neurons. In addition, the amplitude of light-evoked PSCs from VTA neurons (14.4 ± 1.8 pA, n= 8 cells; Supplementary Fig. 7g) and ZI neurons (8.2 ± 1.1 pA, n= 8 cells; Supplementary Fig. 7h) are much smaller than that from SNc neurons (96 ± 15 pA, n= 8 cells). TeNT expression also significantly reduced the amplitude of light-

evoked PSCs from VTA neurons (Supplementary Fig. 7g) and ZI neurons (Supplementary Fig. 7h). These data validated the effectiveness of the dual-AAV strategy to specifically silence the SC-SNc pathway without affecting other tectofugal pathways. The text related to these changes have been highlighted in the revised manuscript (Line 264 - 266).

Supplementary Fig. 7 Control experiment to verify TeNT-mediated synaptic inactivation of the SNc-projecting SC neurons with the dual AAV strategy (a) Schematic diagram showing the dual-AAV strategy to express ChR2-mCherry and TeNT in the SNc-projecting SC neurons of WT mice. (b, c) An example coronal brain section of the SC (b) and a micrograph from the SC (c) showing co-expression of ChR2-mCherry and TeNT in the same SNc-projecting SC neurons. (d) Example micrographs from the SNc (left), VTA (middle), and ZI (right), showing axons of SNc-projecting SC neurons in these brain areas. (e) Schematic diagram showing whole-cell recording of light-evoked postsynaptic currents from the neurons in the SNc, VTA or ZI in acute brain slices. (f-h) Example traces (left) and quantitative analyses (right) showing the effects of TeNT on the amplitude of light-evoked PSCs. Numbers of cells (f-h) are indicated in the graphs. Data in (f-h) are means \pm SEM. Statistical analyses (f-h) were performed using Student t-test (** $P < 0.01$; *** $P < 0.001$; * $P < 0.05$). For the P values, see Table S4. Scale bars are labeled in the graphs.

The TeNT inactivation is irreversible and should be confirmed by optogenetic or chemogenetic inactivation of SC-SNc terminals using the same dual-AAV approach. This could show the effects on both approach behavior and looming response in a reversible manner in the same animals.

We agree with the reviewer that the results of TeNT inactivation should be confirmed by reversible inactivation of the SC-SNc pathway. We address this concern by chemogenetic inactivation of SC-SNc pathway. We injected a mixture of AAV2-retro-Cre and CTB-488 into the SNc, followed by injection of AAV-DIO-hM4Di-mCherry into the SC (Supplementary Fig.9a). CTB-488 was used as an indicator of injection site (Supplementary Fig.9b). This dual-AAV strategy resulted in expression of hM4Di-mCherry in the SNc-projecting SC neurons, which were distributed in the intermediate and deep layers of the SC (Supplementary Fig.9c). We found that chemogenetic inactivation of these neurons by intraperitoneal injection of CNO (1 mg/kg) robustly impaired appetitive locomotion during predatory hunting (Supplementary Fig.9d-9h). However, chemogenetic inactivation of these neurons did not significantly alter defensive locomotion evoked by looming visual stimuli (Supplementary Fig.9i-9l). These data further support that the SC-SNc pathway is selectively required for appetitive locomotion. The text related to these changes have been highlighted in the revised manuscript (Line 307 - 317).

Supplementary Fig. 9 Effects of chemogenetic inactivation of the SC-SNc pathway on appetitive and defensive locomotion. (a) Schematic diagram showing the dual-AAV strategy to express hM4Di-mCherry in SNc-projecting SC neurons. (b) Example coronal brain sections showing the bilateral injection sites of AAV2-retro-Cre mixed with CTB-488. (c) Example coronal brain section showing the distribution of hM4Di-mCherry+ SNc-projecting neurons in the SC. (d) Behavioral ethograms of predatory hunting of example mice without (Ctrl+CNO) and with (hM4Di+CNO) chemogenetic inactivation of SC-SNc pathway. (e-h) Quantitative analyses of hunting behavior showing the effects of chemogenetic inactivation of the SNc-projecting SC neurons on time to capture (e), speed of approach (f), frequency of approach (g), and frequency of predatory attack (h). (i) Time courses of locomotion speed before, during and after looming visual stimuli in mice without (Ctrl+CNO) and with (hM4Di+CNO) chemogenetic inactivation of the SC-SNc pathway. (j-l) Quantitative analyses of locomotion speed before (j), during (k) and after (l) looming visual stimuli, showing chemogenetic inactivation of the SC-SNc pathway had little effect on defensive locomotion. Scale bars are labeled in the graphs (b, c). Numbers of mice are indicated in the graphs (e-l). Data in (e-l) are means \pm SEM. Statistic analyses (e-h, j-l) are performed using Student t-test (** P<0.01). For the P values, see Table S4.

2. *The supplementary movies could be improved by including more information on what is shown instead of requiring going back and forth to the text for context. Information such as the light on/off periods, which are not clear in the video, the frequency of stimulation, the type of mouse and manipulation (TeNT, control, WT, AAV, etc...) would be helpful. Also, if the main parameter is the duration until the last attack, it would be helpful to add the time count explicitly. Perhaps even showing the control and manipulated mice examples side-by-side in the video.*

We thank the reviewer for all these great suggestions. We have added the requested information in the Supplementary Videos of the revised manuscript.

3. *The results shown suggest that there was always a successful and purposeful hunting of the cockroaches by the mice. Was this always the case? Were there cases where there was no interest or no success in the hunting? If so, such cases should be mentioned, especially if they were more prevalent in the SC-SNc pathway inactivation experiments.*

We concur with the reviewer that we have not clearly clarified the success rate of predatory hunting of mice in the original manuscript. In this study, all the mice for experiments exhibited successful and purposeful hunting of the cockroaches. This is the case for both TeNT-mediated inactivation (Fig. 6) and chemogenetic inactivation of the SC-SNc pathway (Supplementary Fig. 9). It is probably because inactivation of the SC-SNc pathway did not apparently attenuate the motivation of predatory attack (Fig.6, g and h). According to the previous studies, the motivation of predatory attack may be a critical factor to determine the success rate of predatory hunting (Zhao et al., 2019; Shang et al., 2019). In the revised manuscript, we have added one sentence to clarify this issue (Line 286 - 288).

4. *The criteria for choosing the 18 cells in Figure 2 are not completely clear. Were there 18 cells that responded with short latency to antidromic light-stim?*

We appreciate the reviewer's concern. Yes, there were 18 units that responded with short latency ($2.7 \text{ ms} \pm 0.4 \text{ ms}$) to antidromic light pulses illuminating on the ChR2-mCherry+ axon terminals in the SNc. In the revised manuscript, we have added one sentence to clarify this issue (Line 215).

Were all of them found to be active spontaneously during non-predatory treadmill walking? If not, how many cells were antidromically activated and excluded from analysis (did not show activity during spontaneous locomotion)? This point should be more explicitly described in the text and figure.

We appreciate the reviewer's concern on our poor clarification in the original manuscript. As shown in Fig.4e and 4f, all of the units (18/18) that were antidromically activated were modulated by treadmill walking. We have added this description in the revised manuscript (Line 220-222).

To further clarify this, we have analyzed the 41 units that were not antidromically activated by light pulses. A large proportion of them (18/41, 44%) were not modulated by treadmill walking, whereas other units were either activated (15/41, 36%) or inhibited (8/41, 20%) at the initiation of locomotion (Supplementary Fig. 5d-5g). The text related to these changes have been highlighted in the revised manuscript (Line 236 - 238).

5. The experiments depicted in Figure 7 are not selective: bilateral chemogenetic silencing of dopamine SNC neurons is not specific, and is expected to cause impairment of movement in general. Was there any change in locomotion following CNO administration? Was there any change in "hunting" behavior? Spontaneous movement? Escape in looming?

We agree with the reviewer that bilateral chemogenetic silencing of SNc dopamine neurons is expected to cause a general impairment of movements. To answer the reviewer's questions, we examined the effects of bilateral chemogenetic silencing of SNc dopamine neurons on various forms of locomotion (Supplementary Fig. 11a-11g). AAV-DIO-hM4Di-mCherry was locally injected into the SNc of DAT-IRES-Cre mice, resulting in localized expression of hM4Di-mCherry in the SNc with very little spillover to the adjacent VTA (Supplementary Fig. 11a). We found that chemogenetic silencing of SNc dopamine neurons by intraperitoneal injection of CNO (1 mg/kg) impaired appetitive locomotion during predatory hunting (Supplementary Fig. 11b-11d), spontaneous locomotion (Supplementary Fig. 11e), and defensive locomotion evoked by looming visual stimuli (Supplementary Fig. 11f and 11g). These data confirm the speculation of the reviewer and indicate that SNc dopamine neurons are required for all three types of locomotion. The text related to these changes have been highlighted in the revised manuscript (Line 354 - 362).

Supplementary Fig. 11 Effects of chemogenetic inactivation of SNC dopamine neurons on appetitive and defensive locomotion in mice. (a) Example coronal brain section showing injection of AAV-DIO-hM4Di-mCherry in the bilateral SNc of DAT-Cre mice (A1), resulting in localized expression of hM4Di-mCherry in the SNc with little leak to the adjacent VTA (A2-A4). (b-d) Quantitative analyses of chemogenetic inactivation of SNC dopamine neurons on the time for prey capture (b), speed of approach (c), and frequency of approach (d) during predatory hunting in mice. (e-g) Quantitative analyses on the effects of chemogenetic inactivation of SNC dopamine neurons on locomotion speed before (e), during (f) and after looming visual stimuli (g). Numbers of mice (b-g) are indicated in the graphs. Data in (b-g) are means \pm SEM. Statistic analyses (b-g) were performed using Student t-test (** $P < 0.01$). For the P values, see Table S4. Scale bars are labeled in the graphs.

The authors show the net effect of Chr2 activation of SC-SNc terminals on such behaviors in the presence and absence of CNO but it is hard to understand how the CNO by itself affected the behavior, especially when there was unspecific silencing of SNc dopaminergic neurons (likely with some spillover to VTA).

We appreciate the reviewer's concern on how the CNO by itself affected the behavior. This is due to our poor clarification in the original manuscript. As we know, this part of work was to examine whether the SC-SNc pathway promotes appetitive locomotion via SNc dopamine neurons. CNO treatment was used for chemogenetic suppression of SNc dopamine neurons that expressed hm4Di-mCherry (Fig. 8c and 8d). We statistically analyzed the effects of CNO treatment on the basal level (laser OFF) of approach speed and approach frequency of mice during predatory hunting (Fig. 8e and 8g). We found that, in comparison with saline treatment, intraperitoneal injection of CNO (1 mg/kg) significantly decreased the baseline speed of approach (Fig. 8e; OFF/Saline: 25 ± 1.9 cm/s; OFF/ CNO: 21 ± 0.7 cm/s; n= 8 mice) and baseline frequency of approach (Fig. 8g; OFF/ Saline: 0.24 ± 0.01 Hz; OFF/CNO: 0.14 ± 0.01 Hz; n= 8 mice). These data indicate that chemogenetic suppression of SNc dopamine neurons impairs basal appetitive locomotion during predatory hunting.

Fig. 8 SC-SNc pathway promotes appetitive locomotion via SNc dopamine neurons. (e) The effect of light stimulation of the SC-SNc pathway on speed of approach during predatory hunting in mice treated with saline (*left*) or CNO (*right*). (f) Chemogenetic suppression of SNc dopamine neurons with CNO significantly attenuated the net effects of light stimulation of the SC-SNc pathway on speed of approach during predatory hunting. (g) The effect of light stimulation of the SC-SNc pathway on frequency of approach during predatory hunting in mice treated with saline (*left*) or CNO (*right*). (h) Chemogenetic suppression of SNc dopamine neurons with CNO significantly attenuated the net effects of light stimulation of the SC-SNc pathway on frequency of approach during predatory hunting. Number of mice (e-h) was indicated in the graphs. Data in (e-h) are means \pm SEM (error bars). Statistical analyses in (e-h) were performed by Student t-tests (n.s. $P > 0.1$; * $P < 0.05$; *** $P < 0.001$). For the P values, see Table S4. Scale bars are indicated in the graphs.

Minor:

1. In figure 7G the units seem different from 7H (net change in frequency of approaches).

We thank the reviewer for pointing out this typo. In the revised manuscript, this typo has been corrected. Please note that Figure 7 in the original manuscript is now Figure 8 in the revised manuscript.

2. The MS should be checked again for grammar, a few examples:

- Line 45: “predator employs” should be “predators employ”*
- Line 46: “How the brain control appetitive” should be “controls”*
- Line 57: “of the SC enable itself to orchestrate” should be “enable it to...”*
- Line 126: “we made single-unit recording from” should be “recordings...”*

We thank the reviewer for pointing out these errors in grammar. In the revised manuscript, the grammar errors in these sentences have been corrected.

Reviewer #3 :

This manuscript by Huang and colleagues investigated the role of SC-SNc pathway in predatory hunting related motor control. In particular, the authors used a combination of anterior and retrograde circuit tracing, optogenetic and chemogenetic activation/silencing of specific pathway, in vivo and ex vivo electrophysiology and behavioral analysis and convincingly showed quite a few interesting findings: First, a specific group of glutamatergic but not GAD positive SC neurons project to SNc. Second, activation of these SC neurons can excite SNc neurons, and evoke DA release in the dorsal striatum. Third, the SC-SNc pathway is particularly important for predatory hunting-related motor behavior. Importantly, the authors showed that inactivation of these projection (by using specific expression of TeNT in SC-SNc projection neurons) only impaired appetitive locomotion, but not defensive (using looming stimulation) or exploratory locomotion. This set of experiments provided a comprehensive understanding of a specific excitatory projection (SC) onto SNc, and how this projection controls a specific motor behavior. The authors use anatomical analysis to determine input from the superior colliculus (SC) to the SNc, and identified these inputs from the SC to be prominent for glutamatergic (vGlut2) neurons, and much sparser for GAD2+ neurons. Using retrograde tracing, they describe the existence of different SC neurons projecting to VTA, ZI and SNc. Even though the authors used non-specific method (CTB), not DA neuron specific (for example, Rab1 virus approach), the combination of anterior, retrograde and slice physiology approaches alleviated such concern. The in vivo electrophysiology recordings are convincing, and the analysis is thorough. Recent studies have shown that dopamine neurons receive convergent synaptic inputs from broad brain regions. The current work sets up a great example for teasing out how different synaptic inputs contribute to the locomotion control in different behavioral contexts. The current study used technologically innovative approaches, the results are clear and nicely presented. The major conclusion is supported by the experimental results. As dopamine neurons have been shown to be tightly related to locomotion control and its importance in neurological diseases such as Parkinson's disease, this work would be of interest to a broad neuroscience audience. Overall, the current study is timely, and highly significant. There are no major concerns regarding the experimental design, data interpretation or conclusion. However, I do have a few suggestions below, which can further improve the manuscript.

We thank the reviewer for these positive comments and careful evaluation of our manuscript. Please see our new experimental data below to address the concerns in the specific comments.

1. To better determine the functional role of SC-SNc pathway in appetitive locomotion, the authors need to compare the SC-SNc pathway with other excitatory afferents to the SNc, for example, the M1-SNc pathway.

We agree with the reviewer that it is essential to compare the SC-SNc pathway with other excitatory afferents to the SNc. To address this concern, we examined whether activation of the M1-SNc pathway increased locomotion speed of mice. AAV-ChR2-mCherry was injected into the M1 of WT mice, followed by optical fibers implanted above the SNc. Surprisingly, we found that photostimulation of ChR2-mCherry+ axon terminals in the SNc did not increase locomotion speed of mice (Supplementary Fig. 10, d-f). These data underscored the specificity of the SC-SNc

pathway to promote locomotion in mice. The text related to these changes have been highlighted in the revised manuscript (Line 327 - 330).

Supplementary Fig. 10 Activation of M1-SNc pathway did not increase locomotion speed of mice. (d, e) AAV- ChR2-mCherry was injected into the primary motor cortex (M1) of *WT* mice (d), followed by an optical fiber implanted above the ChR2-mCherry+ axon terminals in the SNc (e). **(f)** Quantitative analyses of locomotion speed without (OFF) and with (ON) light stimulation of M1-SNc pathway. Number of mice (f) was indicated in the graphs. Data in (f) are means \pm SEM (error bars). Statistical analyses in (f) were performed by Student t-tests (n.s. $P > 0.1$). For the P values, see Table S4. Scale bars are indicated in the graphs.

2. The authors showed that activation of SC-SNc pathway could evoke dopamine release in the dorsal striatum, as evidence by pulsed GRAB-DA signals that were suppressed by Haldol in a dose-dependent manner. Although these data are strong, it would be more convincing if the author can provide more physiological data. For example, are these GRAB-DA signals modulated by locomotion, on a treadmill or during appetitive locomotion of predatory hunting?

We agree with the reviewer that it is essential to examine dopamine release in a more physiological condition. To address this concern, we examined whether dopamine release occurs during treadmill locomotion. In head-fixed mice walking on a cylindrical treadmill, the fluorescence of GRAB-DA sensor was robustly modulated by locomotion (Supplementary Fig. 4a and 4b). When we aligned the GRAB-DA signals with the initiation of locomotion (Supplementary Fig. 4c), we found that the average GRABDA signal started to rise at $125 \text{ ms} \pm 23 \text{ ms}$ before locomotion initiation ($n=6$ mice; Supplementary Fig. 4d). This control experiment confirmed that the GRAB-DA sensor could reliably report dopamine release in the dorsal striatum. The text related to these changes have been highlighted in the revised manuscript (Line 184 - 192).

Supplementary Fig. 4 GRAB-DA signals from the dorsal striatum. (a) Schematic diagram showing fiber photometry recording of GRABDA signals from dorsal-striatal neurons expressing GRABDA sensor in a head-fixed mouse walking on a cylindrical treadmill. (b) Example traces of GRABDA signal (bottom) aligned with locomotion speed (top). (c, d) Heat-map graphs of individual GRABDA signals (c) and averaged GRABDA response curve (d) aligned with locomotion initiation (Time = 0 sec), showing the temporal relationship between GRABDA signals and locomotion initiation. Numbers of mice are indicated in the graphs (d). Data in (d) are means \pm SEM (error bars). Scale bars are labeled in the graphs.

3. Regarding the specificity of TH-GFP mice to label SNc DA neurons (Fig. S7), the authors should perform better analysis. For example, brain sections containing the SNc at different bregma from anterior to posterior should all be included in the analysis. Such detailed analyses should also be done for GAD2-IRES-Cre/Ai14 mice and vGlut2-IRES-Cre/Ai14 mice.

We thank the reviewer for this suggestion. In the revised manuscript, we have performed a more detailed quantitative analyses to examine the colocalization of TH and fluorescent proteins (TH-GFP, GAD2-tdT, vGlut2-tdT) in the SNc as a function of bregma.

Supplementary Fig. 3 A more detailed quantitative analyses of colocalization of TH with fluorescent markers. (a) Example micrographs showing that TH-GFP+ cells and TH+ cells are largely overlapped in the SNc of TH-GFP mice. (b) Quantitative analyses showing that the TH-GFP+ cells in the SNc at different bregma were predominantly TH+ (left), whereas most TH+ cells were TH-GFP+ (right). (c) Example micrographs showing that GAD2-tdT+ cells and TH+ cells are largely segregated in the SNc of GAD2-IRES-Cre/Ai14 mice. (d) Quantitative analyses showing that the GAD2-tdT+ cells in

the SNc at different bregma were predominantly TH- (*left*), whereas most TH+ cells were GAD2-tdT- (*right*). **(e)** Example micrographs showing that vGlut2-tdT+ cells and TH+ cells are largely segregated in the SNc of *vGlut2-IRES-Cre/Ai14* mice. **(f)** Quantitative analyses showing that the most vGlut2-tdT+ cells in the SNc at different bregma were TH- (*left*), whereas TH+ cells were predominantly vGlut2-tdT- (*right*).

4. The sequence of the presentation. The session on SC-SNc pathway innervates SNc DA neuron (slice physiology) and evoked dopamine release should be introduced before predatory locomotion. This way, the whole story starts from anatomy, to physiology, and end with in vivo and behavior. However, this is reviewer's suggestion. This does not affect the conclusion of the current study.

We thank the reviewer for this great suggestion. In the revised manuscript, we have reorganized the sequence of the figures and text according to this suggestion.

We hope that we have been able to address all the reviewers' concerns. And thank the reviewers for their very helpful and constructive comments.

Reviewers' Comments:

Reviewer #1:

Remarks to the Author:

The authors have comprehensively addressed all the reviewer's comments. I now fully support the publication of this study in Nature Communication.

Reviewer #2:

Remarks to the Author:

The authors have addressed all my comments with a set of new experiments, analysis, figures and text.

Reviewer #3:

Remarks to the Author:

The authors have successfully addressed all the major concerns raised by the reviewers. The revised manuscript has been significantly improved. Congratulations to the authors for an exciting and significant study!

Reviewer #1 (Remarks to the Author):

The authors have comprehensively addressed all the reviewer's comments. I now fully support the publication of this study in Nature Communication.

We Thank the Reviewer for supporting us!

Reviewer #2 (Remarks to the Author):

The authors have addressed all my comments with a set of new experiments, analysis, figures and text.

We Thank the Reviewer for supporting us!

Reviewer #3 (Remarks to the Author):

The authors have successfully addressed all the major concerns raised by the reviewers.

The revised manuscript has been significantly improved. Congratulations to the authors for an exciting and significant study!

We Thank the Reviewer for supporting us!